# Modelling seasonal meltwater forcing of the velocity of land-terminating margins of the Greenland Ice Sheet

Conrad P. Koziol[1,2] and Neil Arnold[1]

[1]Scott Polar Research Institute, Cambridge U.K.
[2]now at the University of Edinburgh

*Correspondence to:* Conrad P. Koziol (ckoziol@gmail.com)

**Abstract.** Surface runoff at the margin of the Greenland Ice Sheet (GrIS) drains to the ice-sheet bed leading to enhanced summer ice flow. Ice velocities show a pattern of early summer acceleration followed by mid-summer deceleration, due to evolution of the subglacial hydrology system in response to meltwater forcing. Modelling the integrated hydrological - ice dynamics system to reproduce measured velocities at the ice margin remains a key challenge for validating the present understanding of the system, and constraining the impact of increasing surface runoff rates on dynamic ice mass loss from the Greenland Ice Sheet. Here we show that a multi-component model incorporating supraglacial, subglacial, and ice dynamic components applied to a land-terminating catchment in western Greenland produces modeled velocities which are in reasonable agreement with those observed in GPS records for three melt seasons of varying melt intensities. This provides numerical support for the hypothesis that the subglacial system develops analogously to alpine glaciers, and supports recent model formulations capturing the transition between distributed and channelized states. The model shows the growth of efficient conduit-based drainage up-glacier from the ice sheet margin, which develops more extensively, and further inland, as melt intensity increases. This suggests current trends of decadal timescale slow-down of ice velocities in the ablation zone may continue in the near future. The model results also show a strong scaling between average summer velocities and melt season intensity, particularly in the upper ablation area. Assuming winter velocities are not impacted by channelization, our model suggests an upper bound of a 25% increase in annual surface velocities as surface melt increases to 4x present levels.

## 1 Introduction

Surface meltwater draining into the subglacial system drives seasonal acceleration of ice velocities at land-terminating sectors of the Greenland Ice Sheet margin. It may also be an important factor for seasonal accerlation of marine-terminating sectors (Howat et al., 2010; Sole et al., 2011; Moon et al., 2014). Increased water pressures reduce basal drag by decreasing ice-bed coupling, leading to faster ice flow. Early in the summer, surface runoff drains into an inefficient hydrological system, elevating water pressures and accelerating ice flow (Bartholomew et al., 2011; Fitzpatrick et al., 2013; Sundal et al., 2011). As the melt season progresses, a channelized system that efficiently drains water develops. This reduces water pressures and leads to a late summer deceleration (Bartholomew et al., 2010; Chandler et al., 2013; Cowton et al., 2013; Schoof, 2010). Understanding the impact of increased surface melting (Hanna et al., 2013; van den Broeke et al., 2009) on the spatial and temporal evolution of

basal hydrology is important for constraining the GrIS's future evolution. If increased summer melt intensity drives faster mean annual velocities, than a positive feedback between surface melt and ice flow would contribute to mass loss from the GrIS in a warming climate (Zwally et al., 2002). Faster ice flow would draw ice down to lower elevations, where the melting is greater, which in turn drives faster ice flow.

Observations do not show a simple relationship between surface runoff and ice velocities however. Decadal-timescale observations in Southwest Greenland of land terminating sectors show mean annual velocities decreasing in the ablation zone (Stevens et al., 2016; Tedstone et al., 2015; van de Wal et al., 2015). However reported correlation between summer melt intensity and mean annual ice velocities from these studies are either slightly negative, or nonexistent. In the accumulation zone, decadal-timescale measurements are sparse and the data inconclusive about velocity trends (Doyle et al., 2014; van de Wal et al., 2015). Measurements on a daily timescale in the ablation zone show that increased melt intensity can lead to faster ice flow early in the summer. However, the impact of increased ice motion early in the summer on the average annual velocity can be offset by an earlier onset of channelization and corresponding deceleration as the melt season progresses (Sundal et al., 2011; van de Wal et al., 2015). Increases in channelization extent may also lead to slower mean winter flow, due to more extensive drainage of the subglacial system leading to lower water pressures during winter (Sole et al., 2013). As melt season intensity continues to increase, it remains unclear how ice velocities will be forced by water input at higher elevations where ice thickness is greater, and whether patterns of water input at higher elevations will change (Leeson et al., 2015; Poinar et al., 2015; Cooley and Christoffersen, 2017).

Numerical models can provide insight into the hydrological processes driving faster summer flow. Recent subglacial hydrology models have progressed to simultaneously incorporating both distributed and efficient systems, explicitly treating the interaction between the two (Hoffman and Price, 2014; de Fleurian et al., 2016; Hewitt, 2013; Pimentel and Flowers, 2010; Schoof, 2010; Werder et al., 2013). Current models can reproduce the observed upglacier development of the efficient system through the melt season. When coupled to an ice sheet model, the results broadly reproduce the observed velocity patterns of the GrIS margin (Hewitt, 2013; Pimentel and Flowers, 2010). However, recent hydrological models coupled to ice flow models have not been applied to real domains of the GrIS to model large-scale behaviour of the ice-margin during the summer melt season. Rather, applications to real domains of the GrIS for the summer melt season have either omitted ice flow (Banwell et al., 2016; de Fleurian et al., 2016), used a simplified hydrological model coupled to ice flow (Bougamont et al., 2014; Colgan et al., 2012), or focused on a small domain (Hoffman et al., 2016). Coupling recent hydrological models with ice sheet models allows for important feedback between the distributed system and ice velocities (Bartholomaus et al., 2011; Hoffman and Price, 2014), and allows explicit comparison between GPS velocities and model output. Comparisons to surface ice velocity measurements are an important means for validating subglacial hydrological models, and provide a method for constraining poorly understood aspects of subglacial hydrology (see review by Flowers, 2015). Present challenges in applying coupled ice dynamics/hydrology models to the GrIS margin for modelling seasonal evolution include: the values of parameters; the form of the sliding law which relates water pressures to basal drag; and whether the models presently include the necessary elements. Additionally, modelling surface hydrological input to drive the subglacial hydrology model is in itself a challenge. A variety of methods have been employed, incorporating different drainage elements (e.g. Banwell et al., 2016; Bougamont et al., 2014;

de Fleurian et al., 2016). However, no model including drainage via all of crevasses, moulins, and lake hydrofracture has been used to force a subglacial hydrology model to date.

This paper aims to model summer ice flow in the land-terminating Russell Glacier area of western Greenland for three contrasting melt seasons using a multicomponent model approach similar to the previous work of Arnold et al. (1998) and Flowers and Clarke (2002) on alpine glaciers. The model is then used to test the ice sheet response to higher melt input. A coupled hydrology-ice flow model is produced by integrating a subglacial hydrology model (Hewitt, 2013) with the ice flow model of Koziol and Arnold (2017). This coupled model is driven by surface input from the surface hydrology-lake filling model ("SRLF" model) from Koziol et al. (2017), and initiated using the inversions from Koziol and Arnold (2017). The Russell Glacier area is selected as a study site to take advantage of the numerous observations available. These observations include radar flight lines constraining bed topography (Morlighem et al., 2015), meteorological data constraining climatic input (Noël et al., 2015), and GPS data (Tedstone and Neinow, 2017) which provide a calibration and validation data set for model output.

## 2 Methods

The methods section begins with a description of the Russell Glacier study area and the data sets used. The study site is presented first so that the domain can be referred to when describing the boundary conditions applied in the models. Each individual model is then briefly described, before detailing how the models are linked. The coupled ice flow/subglacial hydrology model is referred to as the 'integrated model' for simplicity. Finally, the modelling workflow is described.

### 2.1 Study Area and Datasets

The Russell Glacier area is a land-terminating sector of the GrIS in Southwest Greenland. The study area boundaries for the SRLF model and the integrated model are shown in Fig. 1. The domain of the SRLF runs is selected to be larger than the integrated model domain to minimize the impact of boundary conditions. A 6 km buffer is used at the northern and southern boundaries of the SRLF domain, based on the reported internally drained catchments by Yang and Smith (2016). The SRLF domain extends 8.5 km to east of the integrated model study site to capture as much higher elevation melting as possible. The domain of the SRLF model is discretized at a 90 m resolution, while the domain of the integrated model is discretized at a 1000 m resolution.

Two different topography datasets are used. The SRLF model is run using surface topography from the GIMP dataset (Howat et al., 2015). The high resolution surface topography is necessary for accurate water routing and so that lake basin topography is accurately preserved. The integrated model is run with surface and bed topography from BedMachine2 (Morlighem et al., 2014, 2015) to take advantage of the mass-conservation methods used to determine basal topography. BedMachine2 provides both topographic datasets at 150 m, although the true resolution is reported as 400 m. This data is reinterpolated to 1000 m resolution.

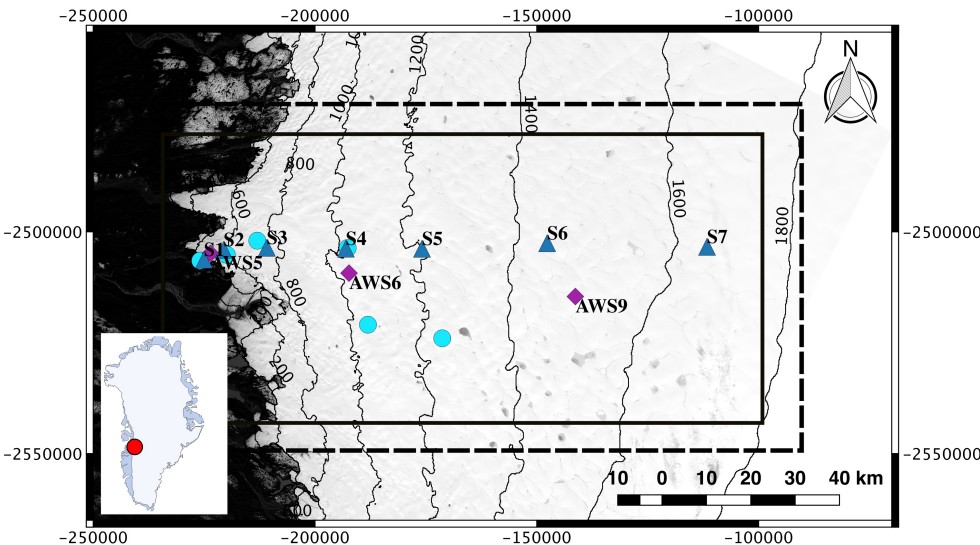

**Figure 1.** Landsat 8 satellite image acquired on 19 August 2013, band 2, showing the Russell Glacier area. Black solid rectangle outlines the study domain for the integrated model, while the black dashed rectangle outlines the SRLF study domain. The blue triangles show the locations of GPS stations (Tedstone and Neinow, 2017). Purple diamonds show the locations of automatic weather stations (van de Wal et al., 2015). Cyan circles show the locations of moulins used as tracer injections sites in Chandler et al. (2013). Inset shows the location in reference to Greenland.

Surface runoff and snow depth data for the SRLF model are provided by RACMO2.3 (Noël et al., 2015). Both runoff and snow depth are bilinearily interpolated from 11 km. Three seasons with contrasting melt volumes are modelled: 2009, 2011, and 2012 (Fig .2). Total melt over the SRLF study domain was $1.2 \cdot 10^{10}$ m$^3$ in 2009, $1.7 \cdot 10^{10}$ m$^3$ in 2011, and $2.1 \cdot 10^{10}$ m$^3$ in 2012. Following Koziol et al. (2017) , we use these three years as representative of summers with average, elevated, and

5  extreme melt intensity respectively.

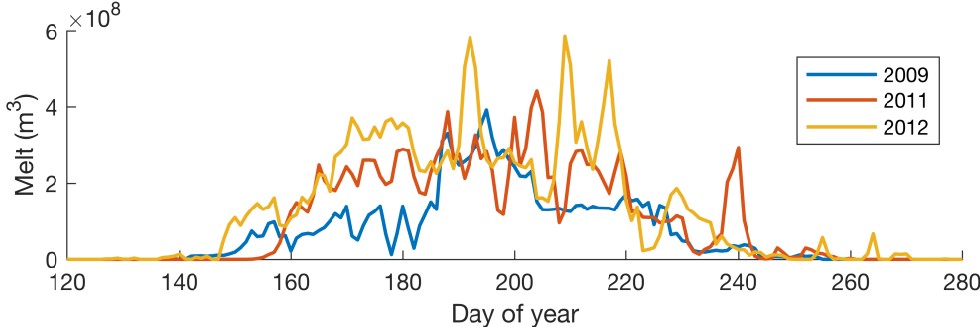

**Figure 2.** Daily surface runoff over the SRLF Russell Glacier study area for three contrasting summer melt seasons.

Mean winter velocities are used for inversions of winter basal boundary conditions (see Section 2.6) and to determine crevasse locations as an input to the SRLF model. Mean winter velocities for 2008-2009 are provided at 500 m resolution by the MEaSUREs Greenland Ice Sheet Velocity Map dataset (Joughin et al., 2010a, b). For the inversion procedure, the winter velocities, along with their associated errors, are reinterpolated to 1000 m. Velocities at 500 m resolution are used to determine

surface stresses, assuming an ice temperature of -5 °C. Crevassed areas are then calculated using a von Mises stress criterion following Clason et al. (2015). A crevassing threshold is selected by comparing the von Mises stress to observed patterns of crevassing in a Landsat 8 image, acquired on 19 August 2013. A threshold value of 145 kPa gave the best visual match.

Moulin locations are specified as input data in the SRLF model. Moulin locations in the Russell Glacier area reported by Yang et al. (2015) are used. These were derived automatically from a Landsat 8 image acquired on 19 August 2013, using an

algorithm which determines where streams are observed to abruptly disappear (Yang et al., 2015). As in Koziol et al. (2017), moulin locations which do not coincide with a stream location calculated by the surface routing algorithm are slightly adjusted, such that they are located on a stream. A small number of moulins from the dataset are deleted, as they were not near a calculated stream, and hence would drain negligible water.

A key validation dataset in the Russell Glacier area is GPS surface velocity measurements for 2009-2012 (Tedstone and

Neinow, 2017). A time series of hourly and daily averaged surface speeds are provided in the dataset. Here, the daily averaged speeds are used for comparison with model results. The locations of GPS stations are shown in Fig. 1.

## 2.2   Supraglacial Hydrology (SRLF)

We use the supraglacial hydrology model (SRLF) from Koziol et al. (2017), run at 90 m resolution. A no-inflow boundary condition is imposed on all boundaries. Water is routed using a DEM of the surface of the ice sheet and a single flow direction

algorithm (Arnold et al., 1998; Tarboron, 1997). Water can enter the subglacial system via drainage into pre-existing moulins or crevasses, and is also allowed to drain off the edge of the domain, or over the western ice margin. Water also collects in depressions in the surface DEM forming lakes. Lakes which are predicted to hydrofracture (Das et al., 2008; Stevens et al., 2015), using a fracture area criterion, drain to the ice-bed interface and create a surface-to-bed connection (treated as a moulin) for the remainder of the melt season. Lakes can also drain over the surface of the ice sheet via "overspill" drainage and

"channelized" drainage. Overspill drainage refers to when water exceeding the capacity of the lake is routed downstream, with no incision of a channel at the lake edge. Channelized drainage refers to when water is routed downstream, but incises a channel at the lake edge, which allows slow lake drainage. Channel incision is modelled following Raymond and Nolan (2000). Overspill and channelized drainage can occur simultaneously if water enters a lake faster than can be evacuated by an existing channel alone.

## 30   2.3   Subglacial Hydrology

We use the subglacial hydrology model presented in Hewitt (2013) and Banwell et al. (2016). Distributed flow occurs through a continuum 'sheet', composed of a cavity sheet component and an elastic sheet component. The latter is included so that during lake hydrofracture events 'hydraulic jacking' is simulated. Channels can form along the edges and diagonals of the rectangular

finite difference mesh. Dissipative heating over an incipient channel width lengthscale provides the initial perturbation for channel initialization. Water input occurs at moulins located at cell nodes, which along with an englacial aquifer, allow for water storage. The model is run at 1000 m resolution. At the ice-margin edge an atmospheric pressure boundary condition is imposed, while the remaining boundaries have a no-flux condition. A concise model description is given here following Hewitt

(2013) and Banwell et al. (2016) to provide context for the parameters used. However, for a detailed description the reader is referred to Hewitt (2013) and Banwell et al. (2016).

Discharge in the continuum sheet is modelled as:

$$\mathbf{q} = -\frac{K_s h^3}{\rho_w g} \nabla \phi \tag{1}$$

where $h(x,y)$ is the thickness of the continuum sheet, $K_s$ is the sheet flux coefficient, $g$ is the acceleration due to gravity, $\rho_w$

is the density of water, and $\phi$ is the hydraulic potential. The hydraulic potential is defined as $\phi(x,y) = \rho_w g b(x,y) + p_w(x,y)$, where $p_w$ is water pressure and $b(x,y)$ is the bed elevation.

The distributed sheet thickness ($h$) is the sum of the thickness of the cavity sheet ($h_{cav}$) and the elastic sheet ($h_{el}$). The cavity sheet evolves according to:

$$\frac{\partial h_{cav}}{\partial t} = \frac{\rho_w}{\rho_i} m + U_b \frac{h_r - h_{cav}}{l_r} - \frac{2A_b}{n^n} h_{cav} |N|^{n-1} N \tag{2}$$

where $\rho_i$ is the density of ice, m is the basal melting rate, $U_b$ is the basal sliding speed, $h_r$ is the bed roughness height scale, $l_r$ is the bed roughness length scale, $A_b$ is the ice creep parameter, $n$ is the exponent from Glen's flow law, and $N(x,y)$ is the effective pressure. The effective pressure is defined as $N = \rho_i g H - p_w$, where $H$ is the ice thickness.

Basal melt rate is given by:

$$m = \frac{G + \tau_{\mathbf{b}} \cdot \mathbf{u_b}}{\rho_w L} \tag{3}$$

where $\tau_{\mathbf{b}} = (\tau_{bx}(x,y), \tau_{by}(x,y))$ is the basal drag, $\mathbf{u_b} = (u_b, v_b) = (u(x,y,b), v(x,y,b))$ is the basal velocity, $G$ is the net conductive flux, defined as the geothermal heat flux minus conductive loss into the ice, and $L$ is latent heat.

The elastic sheet thickness is given by:

$$h_{el} = C_{el} \left[ -N_- + \frac{1}{2} N_0 \max(0, 1 - \frac{N_+}{N_0})^2 \right] \tag{4}$$

where $N_- = \min(N, 0)$, $N_+ = \max(N, 0)$, $C_{el}$ is an elastic compliance, and $N_0$ is a regularization parameter. When effective

pressure is positive, this layer is designed to be zero. As effective pressure approaches zero or is negative, the thickness is determined by the product of the elastic compliance and effective pressure (Banwell et al., 2016).

Discharge in channels is modelled as:

$$Q = -K_c S^{5/4} |\frac{\partial \phi}{\partial r}|^{-\frac{1}{2}} \frac{\partial \phi}{\partial r} \tag{5}$$

where $K_c$ is a turbulent flow coefficient, $S$ is channel cross-section, and $r$ is along channel distance.

The channel cross section evolves according to:

$$\frac{\partial S}{\partial t} = \frac{\rho_w}{\rho_i} M - \frac{2A_b}{n^n} S|N|^{n-1}N \tag{6}$$

where $M$ is the melting rate along the channel wall.

The melting rate along the channel walls is given by:

$$M = \frac{|Q\frac{\partial \phi}{\partial r}| + \lambda_c |q \cdot \nabla \phi|}{\rho_w L} \tag{7}$$

where $\lambda_c$ is an incipient channel width lengthscale (melting of basal ice over this scale contributes to channel initialization).

The equation for mass conservation is:

$$\frac{\partial h}{\partial t} + \nabla \cdot \mathbf{q} + \left[\frac{\partial S}{\partial t} + \frac{\partial Q}{\partial r}\right]\delta(\mathbf{x}_c) + \frac{\partial \Sigma}{\partial t} = m + M\delta(\mathbf{x}_c) + R\delta(\mathbf{x}_m) \tag{8}$$

where $\Sigma$ is englacial storage and $R$ is the supraglacial input rate to moulins. The delta functions apply along channels ($\delta(\mathbf{x}_c)$) and the positions of moulins ($\delta(\mathbf{x}_m)$).

Englacial storage is represented as

$$\Sigma = \sigma \frac{p_w}{\rho_w g} + A_m \frac{p_w}{\rho_w g}\delta(\mathbf{x}_m) \tag{9}$$

where $\sigma$ is englacial void fraction and $A_m$ is moulin cross sectional area.

Model parameters held constant are shown in Table 1. Two parameters of the subglacial hydrology model, $K_s$ and $\sigma$ are the focus of calibration experiments. They were identified in (Hewitt, 2013) and (Banwell et al., 2016) as key parameters determining the morphology of the subglacial hydrology system.

## 2.4 Ice Flow/Inversion

The ice flow model implements the hybrid formulation of the ice sheet stress balance (Arthern et al., 2015; Goldberg, 2011), which can be considered a combination of shallow ice approximation and shallow shelf approximation. The model implicitly accounts for depth varying ice flow, and surface velocities can be explicitly calculated when comparing model output to GPS measurements. This model is similar to the one used in (Hewitt, 2013), except the conservation of momentum equations are a function of depth integrated velocities rather than basal velocities. Parameters for the model are listed in Table 2. A Dirichlet boundary condition is imposed on all lateral domain margins except the ice-margin, where the standard boundary condition based on the continuity of stress is used. A no penetration boundary condition is applied at the edge of the nunatak (Fig. 1). Three sliding laws are implemented:

$$\tau_{\mathbf{b}} = \beta^2 \mathbf{u_b} \tag{10}$$

$$\tau_{\mathbf{b}} = \mu_a N_{sl}^p U_b{}^q \frac{\mathbf{u_b}}{U_b} \tag{11}$$

| Symbol | Constant | Value | Units |
|--------|----------|-------|-------|
| $\rho_w$ | water density | 1000 | $\text{kgm}^{-3}$ |
| $\rho_i$ | ice density | 917 | $\text{kgm}^{-3}$ |
| $g$ | gravitational constant | 9.8 | $\text{ms}^{-2}$ |
| $n$ | exponent in glen's flow law | 3 | |
| $A_b$ | creep parameter | $7 \cdot 10^{-24}$ | $\text{Pa}^{\text{n}}\text{s}^{-1}$ |
| $L$ | latent heat | $3.35 \cdot 10^5$ | $\text{Jkg}^{-1}$ |
| $S_m$ | moulin area | 10 | $\text{m}^2$ |
| $\sigma$ | englacial void fraction | see text | |
| $K_s$ | sheet flux coefficient | see text | $\text{Pa}^{-1}\text{s}^{-1}$ |
| $K_c$ | turbulent flow coefficient | 0.1 | $\text{ms}^{-1}\text{Pa}^{-0.5}$ |
| $\lambda_c$ | incipient channel width | 10 | m |
| $h_r$ | bed roughness height scale | 0.5 | m |
| $l_r$ | bed roughness length scale | 10 | m |
| $h_c$ | critical layer depth | 1 | m |
| $C_{el}$ | elastic compliance | $1.02 \cdot 10^{-5}$ | $\text{mPa}^{-1}$ |
| $A_m$ | moulin cross sectional area | 10 | $\text{m}^2$ |
| $N_0$ | regularization pressure | $10^3$ | Pa |

**Table 1.** Constants used in the subglacial hydrology model during integrated runs in the Russell Glacier area.

| Symbol | Constant | Value | Units |
|--------|----------|-------|-------|
| $A$ | Ice-flow parameter | $7 \cdot 10^{-25}$ | $\text{Pa}^{\text{n}}\text{s}^{-1}$ |
| $A_b$ | Ice-flow parameter for basal ice | $7 \cdot 10^{-24}$ | $\text{Pa}^{\text{n}}\text{s}^{-1}$ |
| $\rho_i$ | Ice density | 917 | $\text{kgm}^{-3}$ |
| $g$ | Gravitational constant | 9.81 | $\text{ms}^{-2}$ |
| $n$ | Exponent in Glen's flow law | 3 | |
| $p$ | Exponent in Budd sliding law | $3^{-1}$ | |
| $q$ | Exponent in Budd sliding law | $3^{-1}$ | |
| $\lambda_b$ | bed roughness scale | 1 | m |
| $t_y$ | Seconds per year | 31536000 | $\text{syr}^{-1}$ |
| $\epsilon$ | viscosity regularization parameter | $1 \cdot 10^{-14}$ | $\text{ms}^{-1}$ |

**Table 2.** Constants used in the ice sheet/inversion model applied to the Russell Glacier Area.

$$\tau_{\mathbf{b}} = \mu_b N_{sl} \left( \frac{U_b}{U_b + \lambda_b A_b N_{sl}^n} \right)^{\frac{1}{n}} \frac{\mathbf{u_b}}{U_b} \tag{12}$$

where $\beta(x,y)$ is a basal drag coefficient, $\mu_a(x,y)$ is a drag coefficient, $p$ and $q$ are positive exponents, $\mu_b(x,y)$ is a limiting roughness slope, $\lambda_b$ is a bed roughness length (Hewitt, 2013). Following Hewitt (2013), negative effective pressures are eliminated by setting $N_{sl} = \max(N, 0)$, and regularized with a small constant ($10^2$ Pa).

The linear sliding law (Eq. 10) is used for the initial inversion of winter mean velocities, while the Budd (Eq. 11) (Budd et al., 1979; Hewitt, 2013) and Schoof (Eq. 12) (Gagliardini et al., 2007; Schoof, 2005) sliding laws are used subsequently. The linear sliding law uses a single parameter to represent all the processes at the ice-bed interface, while the non-linear sliding laws attempt to explicitly incorporate the impact of effective pressure and have a more complex dependence on velocity.

The inversion code used in this paper is described in (Koziol and Arnold, 2017). It is based on automatic differentiation methods (Goldberg and Heimbach, 2013; Heimbach and Bugnion, 2009; Martin and Monnier, 2014), and uses the open source Matlab package AdiGator (Weinstein and Rao, 2011-2016). The gradient of the cost function in this method is equivalent to one calculated using Lagrangian multiplier methods (MacAyeal, 1993; Morlighem et al., 2013) to generate the adjoint model (Heimbach and Bugnion, 2009). The cost function (Eq. 13) has two terms. The first is the weighted square of the differences of measured and predicted velocities. The second is a Tikanov regularization term added for stability.

$$J = \gamma_1 \int_{\Gamma_s} w \cdot (U_{obs} - U_s(\alpha))^2 d\Gamma_s + \gamma_2 \int_{\Gamma_b} (\nabla \alpha \cdot \nabla \alpha) d\Gamma_b \tag{13}$$

where $\gamma_1$ and $\gamma_2$ are scaling factors, $\Gamma_s$ is the surface domain, $\Gamma_b$ is the basal domain, $w(x,y)$ is a weighting function, $U_{obs}(x,y)$ are observed surface ice speeds, $U_s(\alpha, x, y)$ are modelled surface speeds, and $\alpha(x,y)$ is the control parameter. The control parameter depends on the sliding law, and represents $\beta$ in the linear sliding law, $\mu_a$ in the Budd sliding law, and $\mu_b$ in the Schoof sliding law. The inverse of reported errors of surface velocities are used as weights. Modelled surface velocities depend on the control parameter via the sliding law. The inversion procedure minimizes the cost function with respect to the control parameter.

## 2.5 Model Integration

The SRLF model is used to determine supraglacial input rates to the subglacial system. For model integration, we assume that water drainage through surface-to-bed connections is strictly vertical, with no horizontal component. The SRLF model routes water into three different surface-to-bed pathways: moulins, lake hydrofracture, and crevasses. Moulins and surface-to-bed connections from lake hydrofracture are treated identically. All water entering these cells drains into the subglacial hydrology system at that location. However, drainage through crevasse fields requires additional consideration. When water enters a crevassed cell in the SRLF model, no further routing occurs. Since it is unlikely that every crevassed grid cell drains water locally to the ice-bed interface, postprocessing of SRLF output is necessary.

Water drainage through crevasse fields is poorly understood, and the scheme implemented here (Figure 3) is motivated by simplicity. We assume all water in crevasse fields drain to bed, neglecting any refreezing. We also assume that contiguous areas of crevassed cells are hydrologically connected. A crevassed cell in the SRLF model can accumulate water from two sources: 1) local ablation predicted by RACMO2.3; 2) a cell which is on the margin of a crevassed area may have water flowing into it from adjacent non-crevassed cells. Modelling predicts approximately 70% of the water drained by crevasses is intercepted water flow over the ice sheet surface (source 2). This water is concentrated at the points where supraglacial streams intersect the crevasse fields. The model assumes that moulins exist at these points, as high water input would be favourable to nucleating and sustaining moulins. Moulins are only placed in cells with sufficient drainage, determined by a volume threshold. A value of $5 \cdot 10^5$ m$^3$ is selected, corresponding to approximately the median volume drained by moulins outside of lake basins. A lower threshold results in a rapidly increasing number of moulins draining smaller amounts of water. A Voronoi partitioning is then used around the inferred moulins to create internal catchments within the crevasse field. All water in a catchment is assumed to drain into its corresponding moulin. As stated, the SRLF model does not route water within crevasse fields; there is no travel time associated with melt in the internal catchments of crevasses and the moulin.

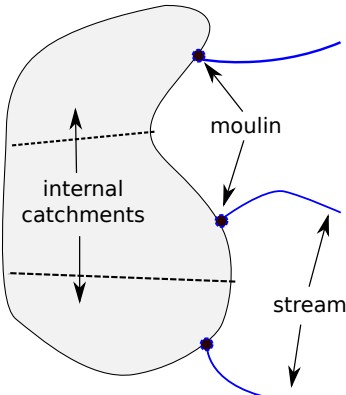

**Figure 3.** Schematic drawing showing the conceptual model of the crevasse drainage implemented. Shaded area indicates a crevasse field. Moulins are assumed to occur where high-flux supraglacial streams intersect the crevasse field. The crevasse field is then partitioned using Voronoi partitioning into internal catchments. All melt which occurs within an internal catchment is assumed to drain into the subglacial system at the corresponding moulin.

The supraglacial model is run independently, to determine a time series and location of water inputs to the base. These are then used as input to the coupled subglacial hydrology/ice flow model. A potential feedback omitted by this approach is the influence of surface velocity on lake hydrofracture. However, the current design and computational requirements of the SRLF model make it impossible to run in a fully coupled manner with the integrated model.

The integration of the subglacial hydrology and ice flow models mirrors that of Hewitt (2013); the subglacial hydrology uses an implicit timestep using the current ice velocity distribution. After the state of the subglacial hydrology model in the next

timestep is calculated, the ice model is called to update ice velocities. At each timestep, the basal melting rate is updated. The geometry of the domain is kept constant for the whole run.

## 2.6 Workflow

Figure 4 shows the workflow for initializing and running the integrated model. The initial step is to perform an inversion using the linear sliding law over the study area (see Koziol and Arnold (2017) for details of inversions, which are done over the same study area but with a resolution of 500 m). All linear inversions are run using mean winter velocities from 2009, the most recent year for which data were available. This inversion provides an initial distribution of basal drag and basal velocities to calculate the basal melt rate (Eq. 3). The subglacial hydrology model is then run for 240 days holding basal velocities fixed, corresponding to a run over a winter season (Sept 1 - April 30). By the end of the run, effective pressures reach an approximate steady state (Koziol and Arnold, 2017). The effective pressures at the end of the subglacial hydrology simulation are then incorporated into an inversion with a non-linear sliding law to determine the background values of the coefficients, $\mu_a$ and $\mu_b$. These sliding law coefficients, the basal water pressures, and the surface runoff input from the SRLF model form the inputs to the integrated model. The integrated model is then run for the summer melt season using the end of winter effective pressures as an initial condition. As stated in Koziol and Arnold (2017), a key assumption of this procedure is that the mean winter velocities are valid both at the beginning and end of the winter season. Although winter velocities are not constant, published GPS records in Southwest Greenland of winter velocities show limited variability (Colgan et al., 2012; van de Wal et al., 2015).

The inversions are run with constant parameters. Both the winter subglacial hydrology run and the subsequent integrated model runs use the same parameters. A parameter search therefore requires performing the inversion using an effective pressure dependent sliding law for each set of parameters tested.

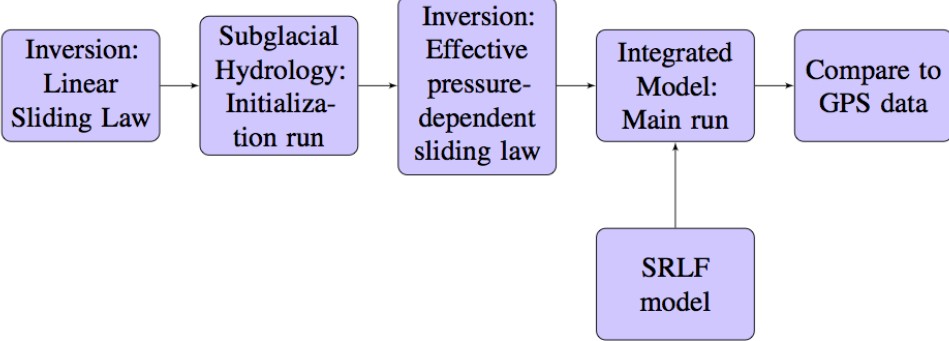

**Figure 4.** Flow chart showing the work flow for initializing and running the integrated model

## 2.7 Simulations

We run five main simulations along with those used in the sensitivity analysis (not shown). Two simulations are run calibrating the model using data and inputs for 2009 and 2011. Another simulation is then run validating the model with data and inputs for

| Pathway | Drainage |
|---|---|
| Crevasses | 24.6% |
| Moulins | 41.8 % |
| Lake Hydrofracture - Lake | 1.3 % |
| Lake Hydrofracture - Moulin | 19.7 % |
| Lake Storage | 0.2 % |
| Remaining Flow | 2.9 % |
| Lateral Outflow | 4.2 % |
| Ice margin | 5.2 % |

**Table 3.** Surface runoff partitioning into different meltwater pathways for the 2009 melt season in the SRLF domain. Water flowing over the western boundary is categorized as 'Ice Margin', while water flow over the lateral boundaries is labelled as 'Lateral Outflow'. 'Remaining Flow' refers to water still flowing over the ice sheet at the end of the model run. Water flowing into crevasses and moulins are in categories 'Crevasses' and 'Moulins' respectively. 'Lake Storage' refers to water in lakes at the end of the simulation. 'Lake Hydrofracture Lake' refers to the water in lakes that is drained by hydrofracture events themselves. 'Lake Hydrofracture Moulin' refers to water drainage into the subsequent surface-to-bed connections from hydrofracture events.

2012. Two potential future melt scenarios are simulated by using 2x and 4x the modelled supraglacial input to the subglacial system for 2011. These are referred to as '2011x2' and '2011x4' respectively. The aim of these scenarios is to investigate potential changes in the behavior of the subglacial system, rather than to model a melt season or reliably predict future ice velocities. Accurate predictions of ice velocities would not only require predicted surface runoff, but also depend on predicting

5 changes in ice sheet topography and predicting the future distribution of supraglacial drainage pathways. Addressing these issues requires careful consideration, and is beyond the scope of this paper.

## 3  Results

### 3.1  Meltwater input partitioning

The majority of supraglacial meltwater drains into the englacial system (Table 3), consistent with observations (Zwally et al.,

10 2002; Smith et al., 2015) and previous modelling (Koziol et al., 2017). Approximately 12% of supraglacial lakes are predicted to hydrofracture. These events drain only a small percentage of surface runoff (1.3%). Most drainage (86.1%) occurs through features modelled as moulins: crevasses, surface-to-bed connections subsequent to lake hydrofracture, and moulins outside of lake basins. Of the water drained by crevasses, approximately 30% is generated locally via ablation in crevassed cells, while 70% is routed into crevasses. Water routing into crevasses is concentrated in a small number of cells, with 50% of the water

15 routed into crevasses entering in only 100 of the 7573 cells forming the perimeter of crevasse fields. Crevasse drainage is concentrated near the ice margin (Figure 5), while drainage into other pathways occurs throughout the study area.

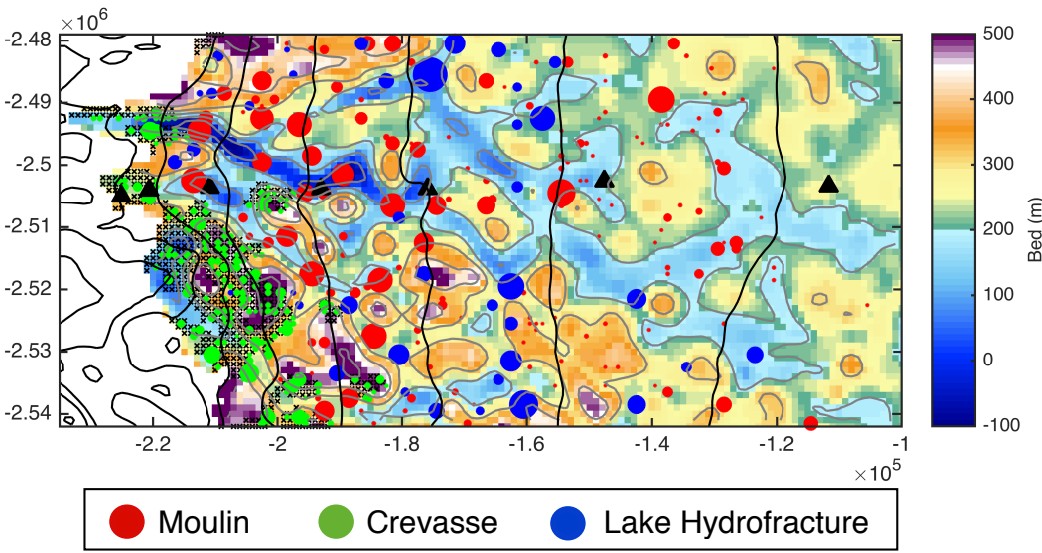

**Figure 5.** Modelled supraglacial input in the Russell Glacier area for the integrated model domain in 2009. Meltwater pathways are denoted by circles of different colors, with red, green, and blue corresponding to moulins, crevasses, and lakes respectively. Circle areas are scaled by volume. GPS stations are shown as black triangles. Hatch marks show grid cells calculated as crevassed. Crevasse inputs appear within hatched areas due to resampling from 90 m to 1000 m resolution. Background is basal topography from BedMachine2 reinterpolated at 1000 m. Light gray contours correspond to 100 m basal contours. Black lines correspond to 200 m surface topography contours at the same elevations as in Figure 1.

## 3.2 Calibration

Model parameters are calibrated by qualitatively comparing modelled velocities to GPS measurements of horizontal surface velocities from 2009 (Fig 6) and 2011(Fig. 7). The focus of the calibration is on parameters identified as key to determining the morphology of the subglacial system, $K_s$ and $\sigma$ (Banwell et al., 2016; Hewitt, 2013). The calibration resulted in the two parameters being assigned spatially heterogeneous distributions. The $\sigma$ field is assigned a background value of $10^{-3}$, with 50% of the cells then randomly set to $10^{-4}$. The $K_s$ field is constructed using a background value of $10^{-2}$ $\mathrm{Pa^{-1}s^{-1}}$, with 15% of the cell nodes randomly assigned a value of $10^{-7}$ $\mathrm{Pa^{-1}s^{-1}}$. Since $K_s$ is defined on the grid, neighboring nodes are averaged in the x and y directions to determine values on edges. At a sheet depth of 0.1 m, a $K_s$ value of $10^{-2}$ results in an effective hydraulic conductivity $K_s h^2$ of $10^{-4}$ $\mathrm{ms^{-1}}$ (Hewitt, 2013). This is at the upper end of values for till, which are inferred to be $10^{-4}$ to $10^{-9}$ $\mathrm{ms^{-1}}$ (Fountain and Walder, 1998). The secondary value of $10^{-7}$ $\mathrm{Pa^{-1}s^{-1}}$ assigned to $K_s$ was selected to give an effective hydraulic conductivity at the opposite end of the spectrum.

The calibration focused on matching the duration and magnitude of speedup events, as the timing of the events is controlled by surface input, which was not sensitivity tested, given we used the same time series of surface runoff for each year from RACMO2.3 in all SRLF runs. The model is calibrated using the Budd sliding law, and the same parameter values are used in

the simulations with the Schoof sliding law. Figures 6 and 7 show model velocities output at noon for the Budd sliding law as representative of the daily average, and daily averages calculated from output at 6 hr intervals for the Schoof sliding law. Sub-daily variability in model output is subdued, except during periods of high velocities in simulations using the Schoof sliding law (see Koziol (2017)). Model output values shown are from the summer immediately following the winter initialization.

There are only minor differences between this model output and from running the model for an additional year and using the output from the second summer. Since surface water input to the subglacial hydrological system is a key driver of ice velocities, surface runoff from RACMO2.3 and nearby surface ablation rates determined at weather stations (van de Wal et al., 2015) are plotted alongside velocities. RACMO2.3 surface runoff forces modelled ice flow, while the weather station ablation rate is taken as representative of the water input driving measured ice velocities. Some caution is necessary comparing the datasets,

since RACMO2.3 accounts for both refreezing of meltwater and precipitation events. Refreezing, however, should only be a small component (van de Wal et al., 2015). An error of 5% is estimated for the calculated daily ablation rates (van de Wal et al., 2015).

The Schoof and Budd sliding laws result in model output of comparable fit to the measured velocities for large segments of the velocity time series. However, during periods of high velocities, the Schoof law can overpredict the magnitude of the

velocity by a factor of 3. Model output with the Schoof sliding law is also observed to have a sharper and higher magnitude summer speedup, as well as a slight increase in velocity variability. Since the Budd sliding law results in an overall better match to the measured velocities, the analysis of the velocity time series in the remainder of the paper focuses on those results.

The model predicts low ice velocities throughout most of the summer melt season at the three GPS sites closest to the margin. In general, measured GPS velocities are also relatively low, except for early season high magnitude variability observed at sites

S1-S3 (Figure 6a-c) in 2009, and at sites S1-S2 in 2011 (Figure 7a-b). This observed variability in the GPS velocities precedes melt predicted by RACMO2.3 and is not reproduced by the model. At sites S1 and S2, modelled velocities show some limited acceleration during the early summer in both 2009 and 2011, but to a lesser extent than in the observed velocities. The fit improves at site S3 for both years, as modelled velocities in 2009 approximate observed velocities (after the early onset of increased summer velocity in the observed data), and in 2011 the model predicts the early summer speedup more effectively.

Modelled velocities at sites S4-S5 (Figure 6d-e and Figure 7d-e) capture the seasonal trend of ice flow, and mirror some of the observed short-term speed-up events. Modelled velocities at S4 match the general flow while diverging from GPS measurements during periods of observed and modelled enhanced flow. In 2011 modelled ice flow shows similar short-term speedup events as the GPS measurements, such as those beginning on days 198 and 237. At site S5, the model does not predict the gradual speedup observed in the GPS velocities. Similar to the GPS measurements, there is a brief period of enhanced flow

in mid-summer, followed by a slowdown. In 2011, however, the model captures the early velocity speedup, and the general trend through the remainder of the summer, including the same speedup events observed at site S4.

Model velocities underpredict the measured velocities at the highest sites. In 2009, measured velocities at site S6 (Figure 6f) show a gradual increase in the first half of the melt season, followed by a gradual decline in the second half. Neither the increase nor decrease in velocity mirrors the weather station ablation rate. In contrast, model velocities are observed to be enhanced in

the middle of summer, mirroring modelled melt. Site S6 (Figure 7f) measurements show faster flow in 2011 than in 2009. The

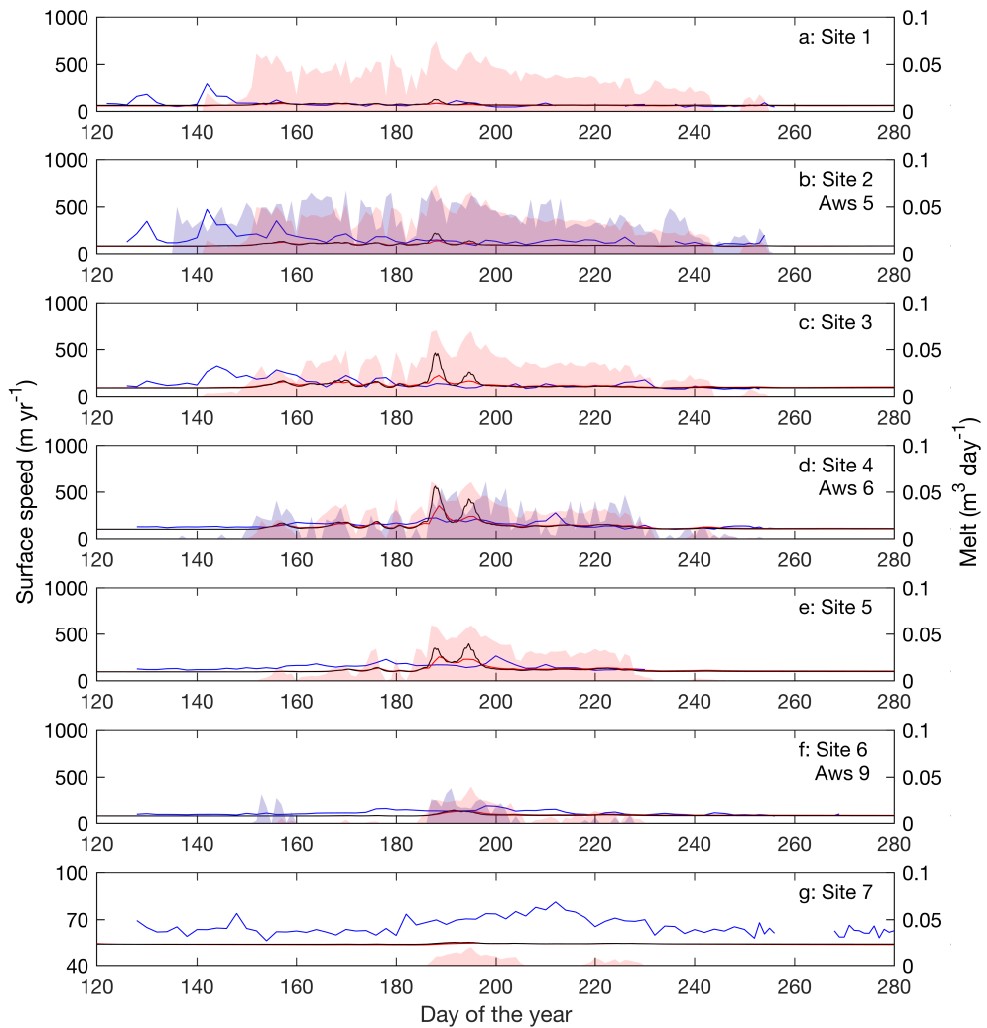

**Figure 6.** Modelled ice velocities plotted against GPS measurements for the 2009 melt season. Daily average horizontal velocity from GPS measurements are plotted in blue. Modelled velocities using the Schoof sliding law and Budd sliding law are plotted in black and red respectively. Daily ablation from weather stations are shown in shaded blue, while RACMO2.3 surface runoff is shown in shaded red. Locations of GPS and weather station sites are shown Figure 1. Weather station ablation rates are plotted at the nearest GPS site.

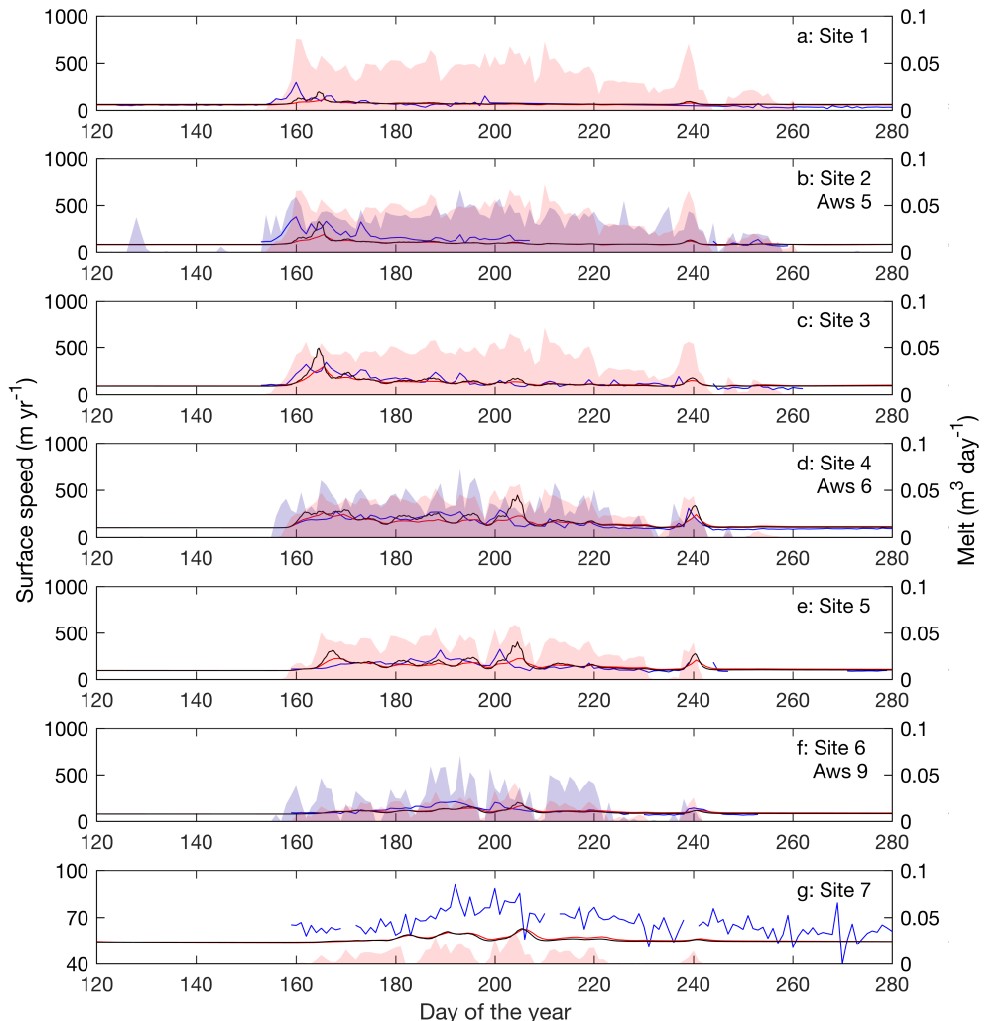

**Figure 7.** Modelled ice velocities plotted against GPS measurements for the 2011 melt season. Daily average horizontal velocity from GPS measurements are plotted in blue. Modelled velocities using the Schoof sliding law and Budd sliding law are plotted in black and red respectively. Daily ablation from weather stations are shown in shaded blue, while RACMO2.3 surface runoff is shown in shaded red. Locations of GPS and weather station sites are shown Figure 1. Weather station ablation rates are plotted at the nearest GPS site.

model velocities match the initial velocity increase observed in GPS velocities, but do not reach the same magnitude. A late summer slowdown is observed in both the modelled and measured velocities, as are short-term increases in velocities at days 200 and 240. At site S7, modelled velocities depart from the winter mean by a few meters per year in both 2009 and 2011 (Figure 6g and Figure 7g). Measurements show an increase on the order of 10-20 $\mathrm{myr}^{-1}$ in both 2009 and 2011.

In summary, the early summer speedup and subsequent mid summer slow-down at mid-elevations are captured. The model is also able to reproduce the pattern of synchronous speedups observed at multiple adjacent GPS stations. Consistent features not captured are early summer variability at low sites, short term variability, and late summer deceleration below the winter mean. Modelled velocities are only observed to flow slower than the winter velocity mean for a period of a few days and by a small magnitude ($< 5$ $\mathrm{myr}^{-1}$).

Ablation rates calculated from automatic weather stations near to sites S2, S4, and S6 are comparable to predicted RACMO2.3 surface runoff. The two data sets show similar magnitude at sites S2 and S4, with higher variability in ablation than runoff. At S6, both ablation and predicted runoff are similar in 2009, while in 2011 ablation is approximately twice the magnitude of surface runoff and has a much higher variability. Qualitatively, model velocities at sites S1-S3 do not correspond with predicted surface runoff, while they do correspond at sites S4-S6. GPS measurements do not in general correspond with the ablation rate

at S2, and only weakly correspond at S4 and S6.

### 3.3 Model Sensitivity

Calibrating the integrated model is an underdetermined problem. Multiple parameters in each cell across the grid are constrained using only seven times series of point GPS data. The parameters selected for the subglacial hydrological model are not unique in giving a qualitatively good fit. Within the parameter space searched, different sets of parameters either enhanced

or dampened the magnitude of the velocity output, or resulted in a velocity signal that significantly diverged from GPS measurements. Extensive sensitivity analysis of the subglacial hydrology component of the integrated model to parameters are conducted in Werder et al. (2013) and Hewitt (2013). In this section we focus on the sensitivity of the model to the setup, to $K_s$, and to $\sigma$.

Drainage through crevasses is poorly constrained, and hence the impact of varying crevasse drainage is tested. Velocities

at the GPS stations are not found to be sensitive to variations in crevasse drainage. The standard value of the moulin volume threshold of $5 \cdot 10^5$ $\mathrm{m}^3$ resulted in crevasse input partitioning into 182 moulins and internal catchments. Changing the threshold value to $10^5$ $\mathrm{m}^3$ and $10^6$ $\mathrm{m}^3$, resulted in 337 and 122 internal catchments respectively. Model output in both scenarios showed negligible changes. Similarly, neglecting water generated over crevasse fields and only using water flowing into the crevasse fields from external streams had little impact on modelled velocities at the GPS stations.

Lake hydrofracture events result in a large volume of water rapidly draining to the base during the event itself, and a surface-to-bed connection which drains water for the remainder of the melt season. The impact of the initial rapid delivery of water on the model behaviour is tested by running a simulation where the water in the lake, when hydrofracture occurs, was not input to the base, but removed from the system. The impact on modelled ice velocities at GPS stations (none of which are near lakes

which undergo hydrofracture) was found to be negligible. The season-long average velocity across the catchment was also not affected.

A heterogeneous sheet flux coefficient field is found to benefit the fit of modelled velocities by increasing the magnitude of the early summer speedup. The results using a constant value of $10^{-2}\,\mathrm{Pa}^{-1}\mathrm{s}^{-1}$ are overall very similar to the calibrated runs, while decreasing the value to $10^{-3}\,\mathrm{Pa}^{-1}\mathrm{s}^{-1}$ leads to model output with prolonged periods of velocities exceeding $400\,\mathrm{myr}^{-1}$. Increasing the number of grid nodes with the sheet flux coefficient assigned a value of $10^{-7}\,\mathrm{Pa}^{-1}\mathrm{s}^{-1}$ from 15% to 30% had a minor impact. Assigning 50% of the grid nodes resulted in a worsening fit early in the summer at site S4, but had little impact at other sites or beyond the initial speedup. Patterning low value nodes into 4x4 patches, randomly seeded at 125 points, was also tested. The number of patches was selected so that if there was no overlap of the patches, 20% of the nodes would be assigned a lower conductivity. Two simulations were conducted with different random locations of patches. One simulation strongly impacted the early summer speedup at site S3 and S4, while the other had a similar effect but on sites S4 and S5.

The GPS records in 2009 and 2011 show differing characteristics, with ice velocities in 2009 showing much less variability and more gradual changes than 2011. The choice of the parameter value setting for englacial storage ($\sigma$) attempts to balance the fit in both years. A better fit was observed with increased englacial storage ($\sigma = 10^{-3}$) for 2009 and decreased englacial storage ($\sigma = 10^{-4}$) in 2011. Increased capacity of englacial storage had the effect of dampening the velocity output. In 2009, this increased the fit of the model predictions by reducing the high velocities observed at sites S3-S5 between days 180 and 200. However, increased englacial storage also reduced the velocity speedups observed in 2011, particularly around day 205, reducing the fit to GPS measurements.

### 3.4 Validation

The integrated model is validated against GPS velocity measurements from 2012 (Fig. 8), using the parameter values determined in the calibration. The pattern of modelled velocities at sites S1 and S2 are similar to those in 2011, with a moderate early velocity speedup followed by a gradual slowdown for the remainder of the summer. Unlike previous years, GPS velocities at site S1 do not exhibit high magnitude velocity variations, improving the match of the modelled velocities. Although the integrated model does not respond strongly to melt input for most of the summer at site S1, it does predict slightly elevated velocities driven by late season input around days 255 and 265, in line with GPS measurements. At site S2, the general pattern of speedup observed in the GPS velocities is mirrored by the modelled velocities. However, the magnitudes are consistently under predicted, particularly those of the short-term high magnitude speedups. The magnitude of modelled velocities improves at site S3 and especially S4, matching the timing but overpredicting the magnitude at S3, and matching both magnitude and timing of events at site S4. Little GPS data is available at sites S5 and S7. Similarly to previous years, model output underpredicts GPS velocities at site S6.

### 3.5 Increased melt scenarios

The high melt scenarios show faster flow early in the summer and higher peak velocities (Fig. 9). After the early summer speedup at sites S1-S3, simulations 2011, 2011x2, and 2011x4 predict broadly similar velocities, although the 2011x2 and

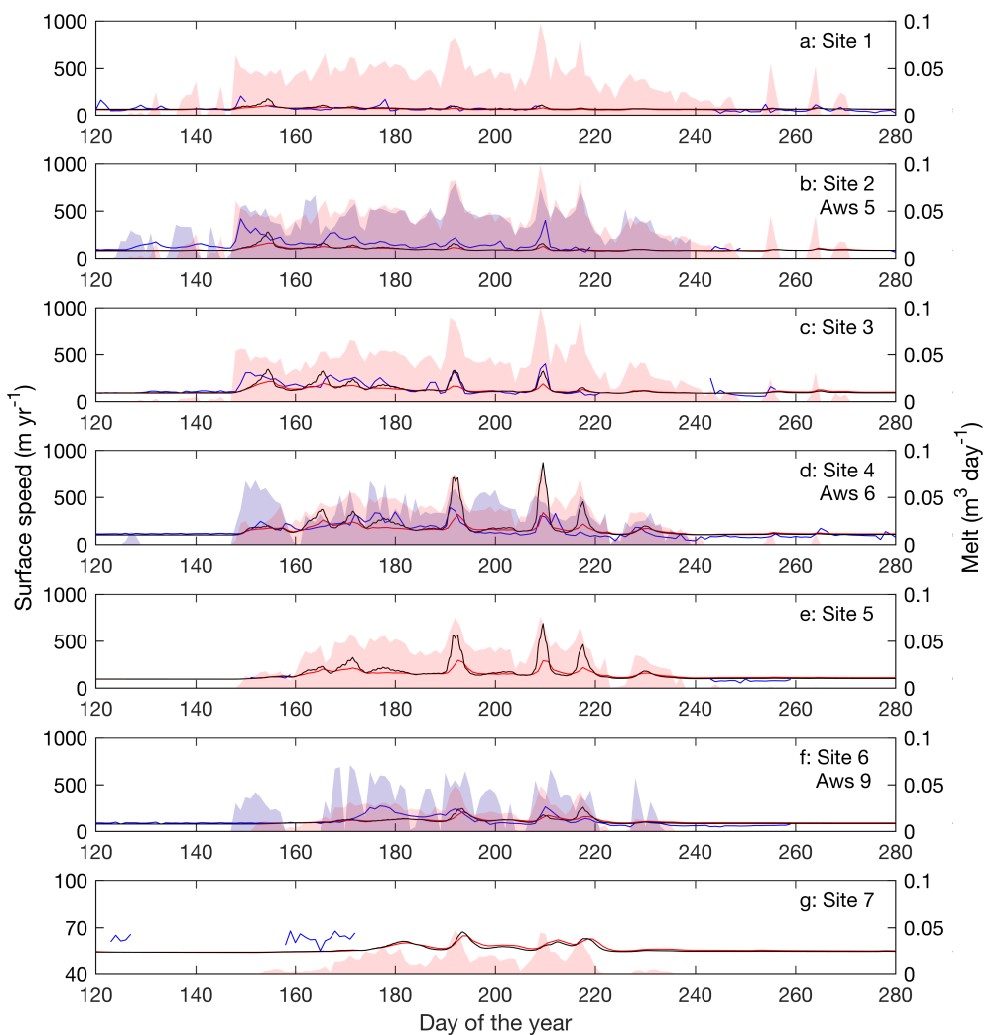

**Figure 8.** Modelled ice velocities plotted against GPS measurements for the 2012 melt season. Daily average horizontal velocity from GPS measurements are plotted in blue. Modelled velocities using the Schoof sliding law and Budd sliding law are plotted in black and red respectively. Daily ablation from weather stations are shown in shaded blue, while RACMO2.3 surface runoff is shown in shaded red. Locations of GPS and weather station sites are shown Figure 1. Weather station ablation rates are plotted at the nearest GPS site.

2011x4 runs show slightly lower values overall; this effect can be seen most clearly at site S3. As elevation increases, modelled velocities in the high melt intensity runs decrease more during slow-flow/low melt periods (e.g. days 210-230 at S4). At sites S4-S6, increased melt intensity results in higher variability of ice flow, with higher peak velocities in the first half of the summer season, but with similar low velocities during low melt periods. The relative increase of peak velocities between 2011x2 and 2011x4 is greater than between 2011 and 2011x2. At these sites, the model predicts broadly similar velocities in all three simulations for the latter half of summer, from days 210 to 238, but at site S4, model velocities are lowest for the 2011x4 scenario, whereas at site S6, modelled velocities increase slightly with greater melt input. Site S5 shows mixed behaviour, with model velocities from simulation 2011x4 higher than the other simulations between days 210 and 222, but lower between days 223 and 236. Between days 210 and 238 at sites S4-S6, model velocities are low and only slightly elevated above their winter values. At site S7, the velocities in the increased melt intensity simulations are faster than 2011 velocities, and with some periods where the 2011x4 scenario generates the slowest velocities (e.g. days 198 and 210), when short-term melt rates drop after a period of higher melt and velocity. Starting at day 238, a late season velocity spike is observed, most strongly at sites S4 and S5, though apparent at other sites. The melt input during this period decreases with elevation, but the impact of this event increases strongly with elevation to sites 4-5, but then decreases at sites S6 and S7.

## 3.6    Average Melt Season Velocities

Melt season averaged modelled velocities at the GPS sites are shown in Figure 10. Average velocities are highest at GPS site S4, and decrease towards the ice margin, and at high elevations. Average velocities increase with melt season intensity at all GPS sites, with a pattern skewed away from the sites closest to the ice margin which show the least sensitivity. As melt season intensity increases, velocities in the upper ablation zone and at the equilibrium line (located at 1500 m elevation (van de Wal et al., 2015), slightly above S6) are predicted to increase the most, scaling with melt (comparing 2011, 2011x2, and 2011x4); site S7, with the lowest melt, shows more limited sensitivity. The pattern observed at the GPS stations is generally reflective of that across the study domain (Figure 11); areas between around 800m and 1400m show the largest increase, but with areas of slower flow predicted to accelerate more than areas of faster flow. Average velocities between 2009 and 2011x4 increase by up to 70%.

## 3.7    Channel Network Morphology/Extent

The development of the channelized system (see supplementary videos) is similar to that observed in previous modelling studies (e.g Banwell et al., 2016; Hewitt, 2013; Werder et al., 2013) and as inferred from observations (Bartholomew et al., 2011; Chandler et al., 2013). Channelization of the hydrological system begins at the margin and develops progressively up ice-sheet. As channelization develops up-ice, the system evolves to an arborescent morphology. The up-ice extent of channelization increases with summer melt intensity (Figure 12). In 2009, channels occur primarily below the 1000 m surface elevation contour. The extent increases past 1100 m, and approaches 1200 m, in 2012. As melt season intensity increases from 2009 to 2012, pockets of channelization at higher elevations are seen. The maximum extent of channelization occurs at approximately the same time in each modelled melt season, and was qualitatively identified to occur between days 220-225 in all three melt

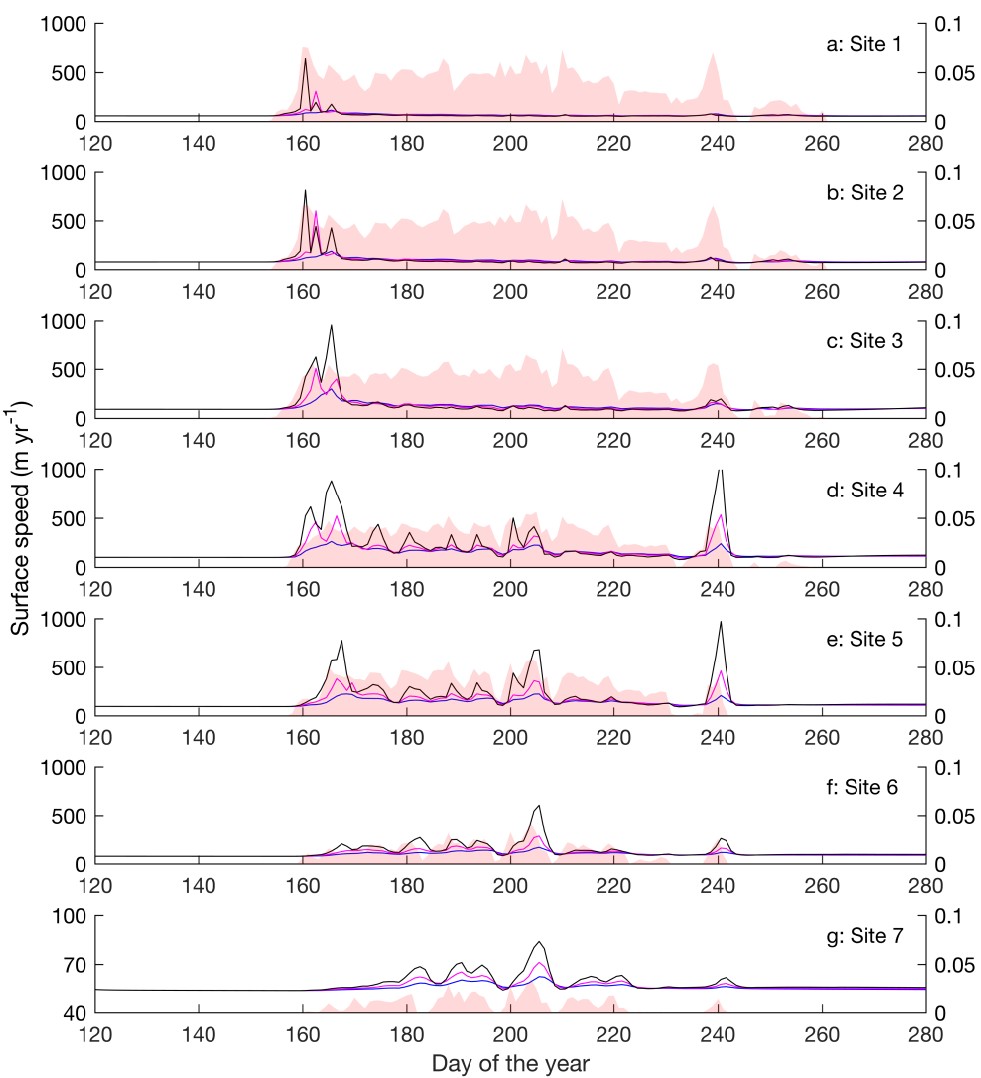

**Figure 9.** Modelled ice velocities using a Budd Sliding law plotted for the 2011 melt season (blue), the 2x melt scenario (magenta), and for the 4x melt scenario (black).

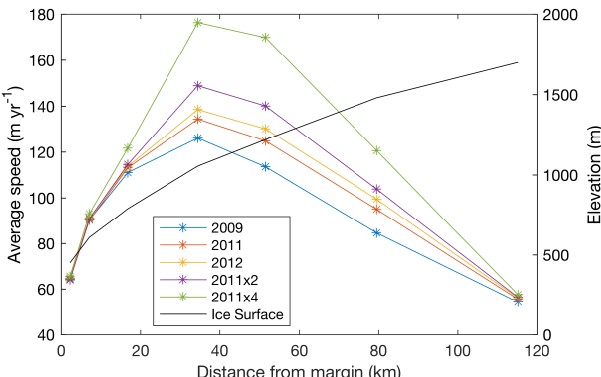

**Figure 10.** Averaged modelled melt season velocities at each of the GPS sites for the different years and future melt scenarios. Ice surface elevation also shown.

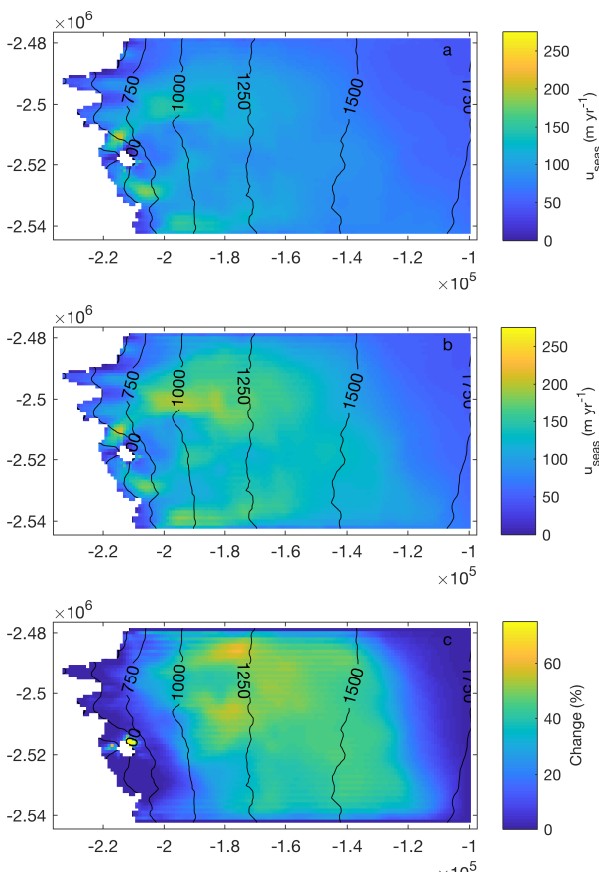

**Figure 11.** a) Map of melt season average velocities for 2009. b) Map of melt season average velocities for 2011x4. c) Change (%) between the melt season average velocities of 2009 and 2011x4.

seasons. Although the extent of channelization varies between 2009, 2011, and 2012, there are no significant differences in the organization of the channelized system. In the future scenario 2011x4 the morphology of the channelized system is similar to that in the modelled melt seasons. However, the extent increases further upstream past 1300 m and approaches 1400 m.

Figure 12 shows the locations of moulins used as tracer injection points in Chandler et al. (2013). Dye-tracing experiments by Chandler et al. (2013) were performed in the summers of 2009 to 2011. Except for moulin IS39, tracers injected into the moulins drained from the subglacial system at an outlet located near moulin L1. Tracers injected into IS39 are reported to drain from an outlet of an adjacent catchment. The channel morphology in the modelled melt season output does not predict a major outlet located near L1, nor that L41 and L57 would drain near L1. However, the model does predict that IS39 is on a different branch of the channelized system. Based on tracer measurements in 2011, Chandler et al. (2013) report that channelization extends to at least L41, but not as far as L57. The modelled channelized system during 2009, 2011, and 2012 is inline with that result.

## 3.8 Distributed and Channelized Discharge

Water flow beneath the ice sheet is modelled to occur in interacting distributed and channelized systems. The discharge in each system follows similar trends for all three modelled melt seasons (Figure 13). In 2009, 2011, and 2012, integrated discharge over the summer melt season in the channelized system is slightly less than half (43%-48%) of the integrated discharge in the distributed system. Modelled discharge begins to increase simultaneously in both systems at the start of the melt season. In 2011 and 2012, discharge in the distributed system rapidly increases in the early melt season. This is followed by a long period with overall high flow but with strong variations. At the end of the melt season, discharge in the distributed system rapidly decreases. In 2009, the early season increase in discharge is less rapid and more prolonged, and discharge peaks before decreasing to a plateau, after which it rapidly decreases. Discharge in the channelized system increases at a much slower rate, and tends to increase until mid-late summer. It mirrors many of the short-time scale variations in the distributed system but with a dampened magnitude. At the end of the melt season, discharge in the distributed system decreases at a higher rate than in the channelized system. In 2011 and 2012, this results in a brief period (after around day 240) in which discharge in channels is higher than in the distributed system. Under the future melt scenario 2011x4, the integrated discharge in the channelized system increases to 77% of the integrated discharge in the distributed system. Early in the melt season, discharge increases in both the channelized system and distributed system simultaneously. Similar to the other melt seasons, discharge in the distributed system increases at a faster rate. However, drainage in the channelized system equals or exceeds that of the distributed system much earlier in the year (Day 200).

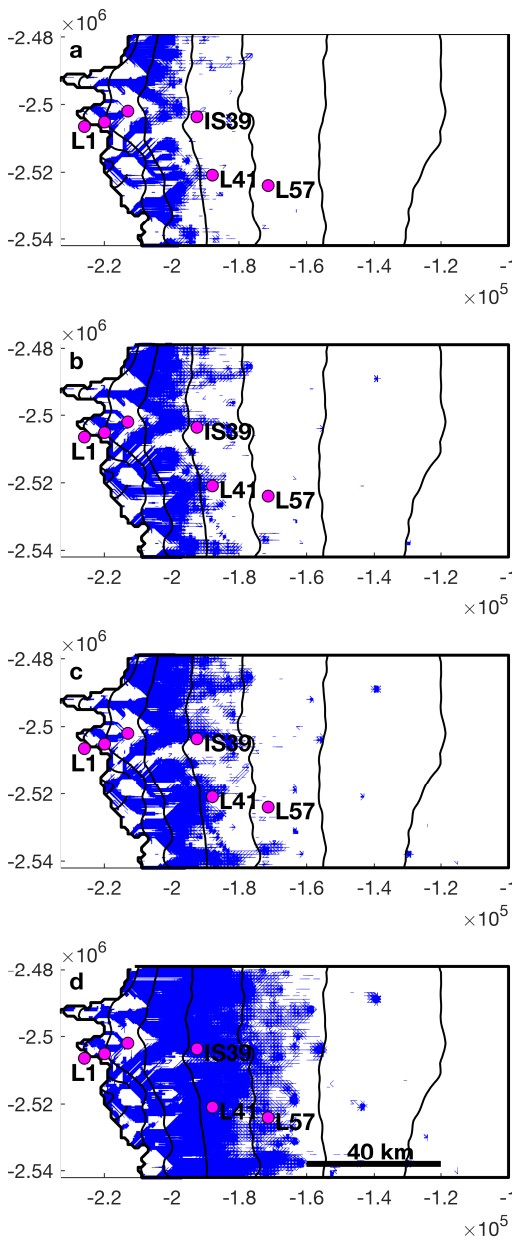

**Figure 12.** Channelized system at maximum extent: a) 2009. b) 2011. c) 2012. d) 2011x4. Moulin locations used as tracer injections sites in Chandler et al. (2013) are shown in purple. Two moulin locations are unlabeled for clarity. Black lines correspond to 200 m surface topography contours at the same elevations as in Figure 1.

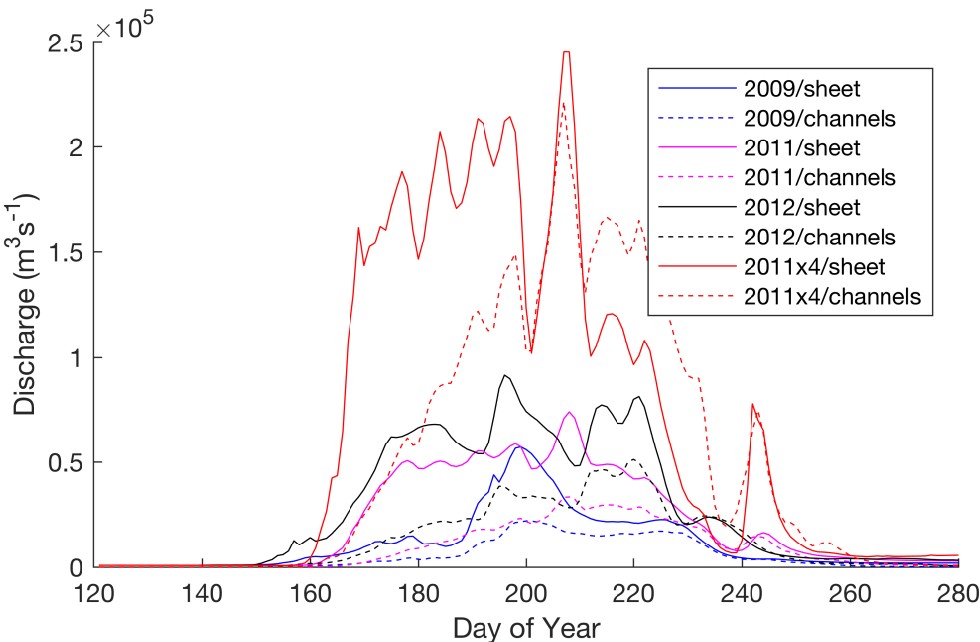

**Figure 13.** Time series of discharge in the distributed ("sheet") and channelized ("channels") system for the 2009, 2011 and 2012 summers, and the 2011x4 future melt scenario.

## 4 Discussion

### 4.1 Model Fit

The modelled velocities are the combined result of five models (RACMO2.3, SRLF, an ice sheet model, the associated adjoint model, subglacial hydrology model) and several datasets. Most parameters used in the models are assigned standard values, with calibrated parameter values for the subglacial hydrology model. The validation simulation affirms the calibrated model, as measured velocities are reproduced to the same qualitative level of fit. Although each model has biases and is limited by assumptions, their combined result reproduces measured ice velocities to a first order. Many of the features observed in the GPS time series are captured in the modelled velocities. This gives confidence that the models and datasets are representative of their respective component. The complexity of the models, and the process, makes assigning a model uncertainty infeasible. It is unclear how to partition the cause of model mismatch between: errors in inputs such as topography, model/theoretical uncertainty such as the form of the sliding law, or computationally imposed limitations such as grid resolution or choice of ice sheet model.

Overall, model velocities are observed to be better at mid-elevation than either at the lowest or highest sites. Model velocities at sites S1-S2 are likely affected by the model not recreating the subglacial water routing inferred by Chandler et al. (2013). A number of factors could contribute to differences in water routing, including errors in topographic data, the spatial distribution

of inputs from crevasse fields, and model assumptions and boundary conditions. In general, thin ice and steep gradients in topography make ice flow and hydrology modelling near the margin difficult. Steep surface gradients may lead to stresses assumed negligible by the hybrid formulation in the ice stress balance. Drainage components in the subglacial hydrology model are formulated in terms of effective pressure, on the implicit assumption that they remain full. Underneath thin ice, or when

there are steep gradients, both channels and cavities could be expected to exist while partially full or empty. The atmospheric pressure boundary condition prescribed at the ice sheet margin in the subglacial hydrology model, may in reality, extend inland for periods in the summer. Additionally, the high velocity spring/early summer events observed in the GPS records occur before any melt is predicted by RACMO2.3. Similar to the study by Bougamont et al. (2014), modelled velocities do not capture this behavior. These velocity events may be the result of internal dynamics of water stored over winter (Schoof et al., 2014), such

as flooding events, that the subglacial hydrology model does not capture, or early season melt which RACMO2.3 does not predict.

    Modelled velocities at sites S6-S7 may be affected by excess capacity in the cavity system due to over prediction of basal ice velocities from the inversion process. The inversion process results in a sliding ratio of approximately 0.8 at the high elevations (Koziol and Arnold, 2017). However, internal deformation can be expected to be dominate over basal sliding so far

inland, suggesting a much lower sliding ratio. Measurements at boreholes in the Paakitsoq region at lower elevations show a sliding ratio of 0.44-0.73 during the winter, increasing episodically to 0.9 during the summer (Ryser et al., 2014). The largest discrepancy between ablation at a weather stations and RACMO2.3 modelled surface runoff occurs at site S6, likely due to RACMO2.3 allowing for refreezing of surface melt. This additional complexity increases the uncertainty in runoff predictions, and surface input to the base may be underestimated at sites S6 and S7.

The development of the subglacial hydrology system is driven by surface runoff input to the bed, and is a key control on velocities across the domain. Measurements in the Russell Glacier area of a single 63.1 $\mathrm{km}^2$ moulin terminating catchment at approximately 1250 m elevation over a 72 hr period found that RACMO2.3 overestimated runoff by approximately 60% (Smith et al., 2017). In general, an average of multiple regional climate models was reported to overpredict surface runoff by +21% to 58% for the catchment (Smith et al., 2017). The impact of any discrepancies between modelled and actual surface

runoff is not necessarily limited to a local temporal and/or local spatial scale. However, to what extent the results of the Smith et al. (2017) study generalize across the model domain is unresolved.

    Spatial maps of modelled velocities show some numerical artifacts. Although these do not appear to have a strong direct impact on the velocities at the GPS stations, numerical artifacts are a cause for concern and should be mitigated in future work. One likely cause is high velocity gradients near the lateral margins due to the Dirichlet boundary conditions. A second

is strong variations in basal drag due to subglacial hydrology likely results in non-negligible horizontal gradients in vertical velocities, contrary to the assumptions of the hybrid formulation. Alternative boundary conditions or numerical schemes to improve convergence may mitigate these effects, but were not pursued further in this study.

## 4.2 Model Sensitivity

Model velocities calculated with the two different sliding laws are comparable during much of the melt season. The timing of events are not affected by the choice of sliding law, and the primary difference observed is the magnitude of velocities during short-term speedup events. The overprediction of speedup during events with the Schoof sliding law suggests adding a regularization constant, such that a minimum basal drag exists. Such a term could reflect the fact that the subglacial hydrological system may not extend throughout a gridcell, or that part of the cell has a weakly connected system with a different water pressure (Hoffman et al., 2016). Simulation results show the Budd sliding law with standard exponent values has practical value in simulations. However, the form and parameters of the sliding law remain uncertain, and the Schoof law has greater theoretical support (Hewitt, 2013).

Calibrating the integrated model is an underdetermined problem, as the number of observations is not sufficient to constrain the parameters in all the models. The calibration therefore focuses on the key parameters of the subglacial hydrology model, while keeping parameters of the ice sheet model and surface hydrology model constant. The calibration was achieved mainly by trial and error, starting with values used in Hewitt (2013). Most model parameters of the integrated model are similar to previous studies applying the subglacial hydrology model (Hewitt, 2013; Banwell et al., 2016). The most significant parameter value difference is the sheet flux coefficient ($K_s$). The primary value of $10^{-2}$ $\mathrm{Pa}^{-1}\mathrm{s}^{-1}$ in our study is greater than the value of $10^{-5}$ $\mathrm{Pa}^{-1}\mathrm{s}^{-1}$ used in Banwell et al. (2016). The parameter values for the model reported in Banwell et al. (2016), which are calibrated against observed water discharge at an outlet in the Paakitsoq region, were found not to reproduce GPS velocity records, as water at mid-high elevations was not effectively evacuated. The difference in parameters suggests that care needs to be taken transferring parameter values between study sites in different areas and at different scales.

The calibrated value for sheet conductivity is at the higher end of inferred values for till (Fountain and Walder, 1998). Although model results are no longer comparable when sheet conductivity decreases by an order of magnitude, model results are resilient to heterogeneity. The simple tests conducted suggest that random heterogeneity in sheet conductivity has a lower impact than larger-scale spatial patterns. Heterogeneity in sheet conductivity could arise from local topography, variable till coverage, and till properties (including deformational history). A constant bed roughness height scale of 0.5 m is used in this paper. However, patterns of bed roughness would also provide a strong control on discharge at the base. Overall, model results suggest it is necessary for the distributed system to be able to sustain a high discharge.

The initial rapid delivery of a large volume of water to the bed during individual lake hydrofracture events is not observed to have a lasting or widespread effect on modelled velocities (in line with observations by Hoffman et al. (2011)). This suggests that lake hydrofracture events in themselves are not a key process in the long term or large scale development of the subglacial hydrological system, as at lower elevations, the numerous conduits and high water input drive channelization, while at higher elevations, a combination of insufficient input and conditions unfavourable for channelization exist. Rather, the primary impact of lake hydrofracture is in opening surface-to-bed connections. The spatial density of such events has been shown to affect the rate of development of channelized drainage (Banwell et al., 2016), and to act as a key mechanism for the creation of moulins

away from crevasse fields or current lake basins (Hoffman et al., 2018), which then drain a significant proportion of the overall surface melt as we find in this study, and previous work (Koziol et al., 2017).

The configuration of internal catchments and moulins which drain crevasses was not found to have a strong impact; neither was eliminating drainage of water generated from ablation in internal catchments within crevasse fields. However, the GPS
sites at which model velocities are compared do not capture spatial heterogeneity of crevasse drainage, which occurs along the length of the ice margin. Hence, the impact may be much stronger at other locations within the study area. However, since model velocities at higher GPS sites were not observed to vary with changes in crevasse drainage, the impact of crevasse drainage should be limited to the margin.

## 4.3  Model Complexity

It is encouraging that the results provide a clear match to many features seen in velocity observations, particularly at the relatively coarse resolution used. However, the models and workflow applied in this paper are characterized by a high degree of complexity. An important consideration is where simplifications can be applied, and where further complexity may be justified.

The use of a higher-order ice sheet model/inversion code should be explored due to increased accuracy in basal velocity calculations. Basal velocities are a key control of the subglacial hydrological system since they determine cavity spacing and
provide an important feedback (Hoffman and Price, 2014). A higher-order model may perform more robustly throughout the study area. Areas where the performance of the hybrid model may be expected to be sub-optimal occur throughout the study area. Such areas are characterized by: high aspect ratio, high variability in basal topography, or low sliding ratio. The ice flow model is also constrained by the assumption of a uniform temperature distribution throughout the ice. Calculating a thermal-mechanical steady state, or alternatively inverting for the structure, would increase accuracy of calculated basal velocities.
Either of these options could be incorporated in the step with the linear inversion at limited cost since this step is only executed once. Importantly, both the use of a higher order ice sheet model and determination of the thermal state can be implemented without adding further assumptions or unconstrained parameters.

The subglacial hydrology model is the least constrained model in the workflow. Many parameters remain unknown and the exploration of its behavior is limited by the parameter space searched. However, a key behavior not replicated is the late
summer/fall slowdown, and subsequent gradual winter acceleration. The integrated model returns to its initial state at the end of summer. This indicates a need for a component of the model which operates on a longer timescale than is currently included. The difficulty in recreating both the smoother 2009 velocity record and the more variable 2011 record also suggests inter-annual variability in the background state of the hydrological system. A model component simulating weakly connected regions of the hydrological system as incorporated in Hoffman et al. (2016) may be key to reproducing these observations. These regions are
conceptualized as parts of the distributed system with a much lower hydraulic connectivity. The connectivity of these regions may be temporally variable.

The SRLF model offers the best opportunity for simplification. To at least a first order, lakes which hydrofracture can be modelled as moulins (in line with observations by Hoffman et al. (2011)). This suggests using the locations of moulins derived from satellite imagery acquired at the end of the melt season as representative of both moulins outside of lake basins, and

hydrofractured lakes. Lake hydrofracture events are observed to result in temporarily faster local flow (Stevens et al., 2015; Tedesco et al., 2013). In order for a model to capture these events, the specific location, timing, and volume of lakes will need to incorporated into the model. Given the ongoing uncertainties around the processes controlling hydrofracture, this suggests that using observational records of lake drainages derived from satellite imagery (as in Bougamont et al. (2014)) to derive hydrofracture input to the ice-bed interface forms a valid strategy for present-day studies, though such an approach would not work for prognostic tests. Crevasses also drain a significant proportion of water, most of which travels over the ice surface into crevasse fields from upstream, rather than being generated locally. The controls on water drainage through crevasses to the ice sheet bed are poorly understood, but may have an important role as the spatial density of water inputs are known to influence the development of the subglacial hydrological system (Banwell et al., 2016). Since moulins and crevasses drain water in a continuous manner with a high spatial density, a simpler surface hydrology scheme approximating input into each drainage pathway from its local catchment may be effective. The output of each catchment into the corresponding drainage pathway may be simplified to two output hydrographs, one for snow-covered and the other for bare-ice conditions. For internal catchments of crevasse fields routing can likely be neglected. This calculation needs only be done once; moulin input at each time step could then be calculated at little computational cost based on total surface runoff and the dominant surface cover in the catchment.

## 4.4 Implications

The existence of channelized and distributed systems beneath the GrIS is inferred indirectly through borehole observations, dye tracing experiments, and patterns of GPS velocities, building on extensive observations and theoretical developments derived from studies on alpine glaciers. The key result of this paper is to provide numerical support for the understanding of subglacial hydrology of the GrIS, based on theories derived from studies of alpine glaciers, as well as support for the explicit description of the model components we include (i.e. the equations used). We show that these theories can quantitatively reproduce measurements to a first order, and in the sense of our validation, predict ice velocities. This builds on previous work which shows that this understanding can be used to reproduce idealized seasonal patterns of ice velocity (Pimentel and Flowers, 2010; Colgan et al., 2012; Hewitt, 2013), as well as effective pressures in line with ice velocities (Werder et al., 2013; de Fleurian et al., 2016).

The timing of velocity variations are controlled by surface input and modulated by subglacial hydrology. At high elevations where channelization is not observed, variations in model velocities track modelled surface runoff closely. GPS velocities, however, do not show the same fidelity to the time series of ablation from automatic weather stations, which are qualitatively more variable than modelled runoff. This suggests dampening of the variability of surface input by the supraglacial and subglacial hydrology, and that variability in daily ablation rates are not simply correlated to faster flow. A quantitative analysis of the two time series may provide better insight into the relationship between surface melt and ice velocities. However ice velocities are driven by the cumulative melt over a larger upstream area from the point of measurement, which may not be well represented by the variability of melt at a single point. At lower elevations, channelization is important in modulating the impact of surface water on ice velocities. The low modelled and observed velocities closer to the ice-margin imply a consistently high effective pressure at the GPS sites, due to the impact of channelization on water pressures and water routing.

Modelling predicts that average summer ice velocities over the melt season will increase with melt season intensity. A similar correlation was observed in GPS records over the upper ablation zone of the Russell Glacier region by van de Wal et al. (2015), but not in GPS records at North Lake, Western Greenland by Stevens et al. (2016). This implies that more intense melt seasons will result in a higher ice flux towards the margin during the summer. Whether this would be compensated for in terms of the average annual ice velocity by decreased ice flux during the winter is unresolved by the model.

Channelization is observed to develop more extensively and further inland as melt intensity increases. This trend is observed in the three modelled melt seasons, and continues into the two future melt scenarios. This suggest that the subglacial hydrological system will continue to drain surface meltwater input in a similar manner as melt intensity increases beyond 2012 levels. Since channelization is thought to result in the observed slowdown in mid-late summer (Tedstone et al., 2015; van de Wal et al., 2015), and is also postulated to result in slowdown in the subsequent winter and spring (Sole et al., 2013; Tedstone et al., 2015; van de Wal et al., 2015), model results suggest increasing summer melt intensity could lead to a more spatially extensive annual velocity slowdown. The slowdown may also become more pronounced in the future as the channelized system is predicted to drain an increased proportion of water, and accesses a larger proportion of the model domain.

However, as channelization increases up-ice in our model, we do not see a marked impact on model velocities. Model velocities at the higher GPS stations in model runs 2011x2 and 2011x4 both show a similar pattern to 2011, with a higher magnitude of ice flow during speedup events. We do not observe a shift in velocity patterns towards that of lower GPS stations, with acceleration early in the melt season transitioning to deceleration in the latter part of the melt season. This suggests that channelization may have a more limited impact on annual velocities in the accumulation zone. The magnitude of any impact is unresolved by our model. In particular, although there are periods when velocities in the 2011x4 run are lower than 2011x2 and 2011, the magnitude of this decrease is bounded. This is a limitation of the model, which is only able to decrease velocities nominally below the initial winter values. This also implies that we are unable to model the winter season accurately.

The key question for the longer term response of the ice sheet to increased melt is whether the potential summer increase in velocity due to increased melt will outweigh any late summer and winter decrease due to the evolution of a more efficient system under higher melt conditions. While observations show long-term decreases in ice velocities in the lower ablation zone (Stevens et al., 2016; Tedstone et al., 2015; van de Wal et al., 2015), this question remains unresolved at higher elevations. Although we cannot directly predict annual velocities with the model presented in this study, we can investigate how annual velocities may change at the limits of winter behaviour.

One limit of winter velocities is that integrated ice flow over the winter decreases faster than integrated ice flow over the summer. This is observed in GPS measurements in the upper ablation zone in the Russell Glacier region by van de Wal et al. (2015). Under this limit, channelization has a similar impact at high elevations as in the ablation zone. Velocity measurements near the vicinity of a lake hydrofracture at approximately 1450 $\mathrm{m}$ elevation suggest that channelization occurs even at high elevations (Bartholomew et al., 2012; Nienow et al., 2017). Winter velocities in this limit will decrease until they hit a lower bound where flow is purely deformational, with no contribution from basal sliding. The maximum increase in mean summer velocities is approximately 60 $\mathrm{myr}^{-1}$, at GPS stations 4 and 5 between the 2009 and 2011x4 melt scenarios. Assuming a winter velocity of 100 $\mathrm{myr}^{-1}$ and an 7 month-long winter, the summer increase predicted by the model would compensate for

a possible reduction in winter velocity to around $60 \mathrm{~myr}^{-1}$. This approaches the lower bound for winter velocity suggested by borehole measurements showing that internal deformation accounts for 25-50% of the total ice velocity in the Paakitsoq region of western Greenland (Ryser et al., 2014). Climate model predictions suggest surface runoff rates quadrupling from present levels by circa 2100 (Shannon et al., 2013; Van Angelen et al., 2013).

5   The second limit occurs if winter velocities at higher elevations are not impacted by channelization, and summer velocities dominate the annual signal. The argument that thick ice and shallow surface slopes inhibit channel growth at high elevations favour this limit of behaviour (Meierbachtol et al., 2013; Dow et al., 2014), as do observations suggesting limited changes in the efficiency of the channelized system (Andrews et al., 2014). Under this scenario, a change of mean summer velocity of $110 \mathrm{~myr}^{-1}$ to $170 \mathrm{~myr}^{-1}$ with winter velocities remaining constant at $100 \mathrm{~myr}^{-1}$ would result in mean annual velocities 10   increasing from $104 \mathrm{~myr}^{-1}$ to $129 \mathrm{~myr}^{-1}$ between the 2009 and 2011x4 simulations. Under this scenario, annual velocities would increase by approximately 25% by circa 2100, when surface runoff is predicted to quadruple.

Interpreting the model velocity output from the future melt scenarios is difficult however, and our bounds should be interpreted cautiously. We do not evolve our model geometry, nor evolve the distribution of surface drainage locations, nor use climate model predictions of surface runoff for the future. As melt season intensity increases, the validity of the initialization 15   and calibration parameters also becomes more uncertain. Further, the model has bias towards capturing short-term speedup events, rather than prolonged slowdowns due to model velocities remaining near or above their winter values. The modelled velocities show higher variability, and a significant increase in the magnitude of short-term speedup events. However, quantifying whether the impact of these events on annual velocity will be compensated for by a corresponding late summer slowdown or by a winter slowdown is beyond the capability of the current model. Model output can be interpreted to suggest that a late 20   summer velocity slowdown compensating for an early summer speedup is less likely at higher elevations. It is not evident, however, whether the suggested upper bound of a 25% increase in annual velocities in this limit would have an impact on the overall mass budget of the ice sheet as great as that from a 4x increase in surface runoff in itself.

The success of the model in recreating features in the measured velocities provides validation for each model component, as well as their integration. The work supports integrating models of high complexity that incorporate a range of processes. 25   Further model refinement and data acquisition should continue to improve the fit between modelled and measured velocities. A key uncertainty in the initialization process was the subglacial hydrology model run during winter, and the subsequent inversion for background basal parameters. Although the process used cannot capture year on year changes, the practical value of the initialization process is implicitly validated through the subsequent fit to measured velocities.

## 5   Conclusions

30   In this paper, we couple multiple models in order to predict summer ice velocities at the southwest margin of GrIS from topographic and climatic input data. These models represent the main components of the ice sheet system: supraglacial hydrology, subglacial hydrology, and ice flow. The key component of the simulations presented in this paper is a coupled hydrology-ice flow model. This integrated model is initialized using a workflow incorporating the adjoint ice flow model, and is forced during

the simulations using surface input from a surface hydrology model. Calibration of the integrated model takes advantage of GPS velocities from two summer melt seasons: 2009 and 2011. The model validation on 2012 GPS data reproduces measured ice velocities to a similar degree as in 2009 and 2011. To a first order, the magnitude and timing of the measured velocities are replicated in modelled velocities at multiple sites.

The success of the multicomponent modeling to recreate summer velocities reflects on the integrity of each individual model and dataset. This work should encourage further model coupling as it suggests that individual components and datasets are robust. However, limitations of the multicomponent model are evident in the model output, particularly that the model velocity does not significantly drop below its initialized winter value. Additional data and theory will be necessary to address these issues. Together, the models also form a quantitative test of the hypothesis proposed by numerous authors (e.g Chandler

et al., 2013; Cowton et al., 2013; Colgan et al., 2011; Hewitt, 2013; Hoffman et al., 2011; Schoof, 2010; van de Wal et al., 2015; Nienow et al., 2017) that the summer acceleration of the GrIS margin is controlled by the evolution of the subglacial hydrological system in a manner analogous to the seasonal speedup of alpine glaciers. The key result of this paper is quantitative support in favour of this hypothesis.

The observed decadal-timescale slowdown at the margin of the GrIS is attributed to increased channelization reducing late

summer and winter water pressures (Stevens et al., 2015; Tedstone et al., 2015; van de Wal et al., 2015), and hence velocities. Our results suggest that the decadal slowdown will continue in the near future, particularly close to the ice margin. However, the model predicts a strong scaling of the average summer velocity with melt season intensity. We investigate the impact of this under two limits. If integrated ice flow over the winter decreases faster than integrated ice flow over the summer at higher elevations, our modelling suggests that annual velocities in the upper ablation zone would begin to increase by around 2100

(when surface runoff is predicted to quadruple from present levels), as predicted summer velocity increases offset likely winter velocity decreases. In the second limit, in which winter velocities remain at present levels while summer velocities increase, our model suggests an upper bound of a 25% increase in annual velocities by around 2100.

*Competing interests.*   The authors declare no competing interests

*Acknowledgements.*   We would like to thank M. Morlighem and I. Joughin for the BedMachine2 and MEaSUREs datasets, and M.R. van den

Broeke and B. Noel for providing RACMO2.3 data. CPK would like to acknowledge R. Arthern for guidance on writing an ice sheet model and inversion code, and B. Minchew, P. Christoffersen, and D. Goldberg for thoughtful discussions. Both authors thank I. Hewitt for sharing model code, the two referees for their careful and constructive reviews, as well as the editor A. Vieli. CPK was funded through St. John's College, Cambridge, and in part by UK Natural Environment Research Council Grant NE/M003590/1.

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
