# Peer review of "Modelling seasonal meltwater forcing of the velocity of land-terminating margins of the Greenland Ice Sheet"

_The Cryosphere, 2017_

## Referee Comment (RC1) · Anonymous Referee #1 · 29 Nov 2017

This is a comprehensive study that models seasonal ice surface velocities at a land-terminating glacier catchment in Greenland. The work pulls together other published component studies by the authors, including a model of supraglacial meltwater pathways that drain through moulins into a subglacial hydrology model, which is coupled to an ice-flow model whose sliding parameters have been determined from an inversion using observed winter velocities. It is a challenging task to bring together many modelling and observational components and the results show some success. The paper raises many interesting points, including a comparison of results using the Weertman and Schoof sliding laws.

The differences between GPS velocities and modelled velocities highlight areas where future improvements can be made in model development. For example, it is interesting

to see some similar challenges in Pimentel et al., Ann. Glac., 2017, e.g. difficulty in capturing early/pre-season speed-up and tendency in the model (with Schoof sliding law) for short-term speed-up events rather than more prolonged enhanced velocities.

In summary I do not have any major problems with the work and think it an interesting and valuable contribution that should be published. I list some minor issues and points of clarification that I would like to see addressed/corrected before final publication.

List of comments:

Title: The work is on one particular catchment of the Greenland Ice Sheet, rather than the entire Ice Sheet, this is not clear in the title.

line 5: The acronym 'GrIS' is not defined

lines 8-10: Do we need further evidence to support that subglacial develop analogous to alpine glaciers and that models need distributed and channelized? This seems quite well established now in the field. Do you want this in the abstract as the key finding of this paper? At the very least 'support' and 'supports' should be changed to 'further support' and 'further supports'.

line 11: 'slow down' of what?

line 16: The acronym 'GrIS' is not defined

line 16: 'margin' should be land-terminating margin, because obviously marine terminating has other additional important influences.

lines 12-13: Poor sentence, needs rewriting to make clearer.

lines 21-23: Presumably just talking about GrIS here.

line 30: typo ". . . to the drive the . . ."

line 32: I'm not quite sure what you mean by "full spectrum of supraglacial drainage pathways"

line 34: How is it similar? You don't seem to have components in the same sense as Arnold et al (1998) and Flowers & Clarke (2002)!?

line 29: typo ". . . with with . . ."

line 30: A better reference is needed for context of Greenland melt intensity

line 31: ". . . to determine crevasse locations . . ." How? Perhaps reference later section at end of sentence.

Figure 1: typo "loactions"

line 5: "A threshold value of 145kPa is selected as optimal." Optimal in the sense that it provided the best match to Landsat 8 image?

line 10: "A small number of moulins . . ." could quantify percentage.

line 18 & 20: ? in citations

line 21 & throughout: "surface to bed" "surface-to-bed"

line 6: "K_s" is "K" in equation (1) and table 1, so I think this should be "K" for hydraulic conductivity, whereas you have "K_s" to denote "sheet flux coefficient" in the table. However, this K_s does not appear in the text/equations! Hydraulic conductivity is given a value of 2 in the table, which does not match the description in the text!

line 5-7: rho_w and p_w are not defined (rho_w is defined later on line 13).

line 19 & throughout: use \max and \min in latex

line 19-20: N_0 a value is not provided in Table 1.

line 6: Presumably "R" is a rate and is a source term from the SRLF?

line 10: You take the derivative of Eqn (9) wrt time as a term in Eqn (8). But is Eqn (9) constant!? Or is A_m a function of time?

line 12: "The englacial storage parameter ..." is this sigma – the "englacial void fraction"!?

lines 12-17: units are missing when giving values of conductivity.

line 16: "khˆ2" should be capital K "Khˆ2"

line 7: "... a small regularization constant" – so this means the zero in N_+ = max(N,0), is not actually zero!? Is this constant different from "N_0" used earlier in equation (4)? What value do you use for this regularization constant?

line 16: "... weighted square of the difference of squares ..."!? This doesn't match equation (13) so I assume this should be square of the differences, rather than difference of squares!?

line 19: please provide your chosen values for gamma_1 and gamma_2

line 18-22: How was the weighting function chosen?

line 18-22: U_obs(x,y) – are these just the 7 GPS sites? So the weighting function is 1 at these locations and zero elsewhere!? Ok, I just checked back to Section 2.1, so U_obs is the winter mean from MEaSURE. And the weights are from the provided error estimates from MEaSURE!?

line 18-22: Why isn't "U_s" also written "U_s(x,y)"?

line 18-22: It would be clearer if you explicitly said that U_s is a function of \alpha and you seek to minimize J with respect to \alpha. If I have understood correctly?

line 24: An odd/abrupt start to this subsection

line 25: There is englacial storage in the subglacial model, but water from the SRLF is not directed to this storage – correct!?

line 27 "All water entering these cells drains directly to the bed" So water is not stored in the moulins – how does this relate to Eqn (9)!?

line 29: "... the water is removed from the model ..." do you actually mean the water vanishes!? Because on page 10 line 2 you say "... all water in crevasse fields drain to bed ...". If I understand correctly the water in the crevassed grid cells ends up draining into a nearby moulin where it then enters the bed.

line 29: "... and no further routing occurs." It is instantly routed to the corresponding moulin – if I understand your next paragraph correctly!?

line 11: "veronoi" "Veronoi"

Figure 3: It may be helpful to somehow illustrate the crevasses (crevasse field). You could possibly use hatched grid cells, like in Figure 6. I suppose this is implemented in a finite difference grid, so the curved lines of the internal catchment would actually be straight edged.

line 8: "The effective pressures at the end of the ... simulation ..." Presumably they've stabilized by then?

line 10: "coefficients." State the coefficients, i.e. "coefficients, mu_a and mu_b."

line 10: "... the basal water pressures, ... form inputs ..." how are the basal water pressures used? Initial conditions of the subglacial summer simulations!?

line 16: "A parameter search ...". I'm not sure what you are getting at with this sentence.

line 18: "... sensitivity analysis (not shown)." Section 3.3!?

Section 3.1: What percentage of your lakes hydrofracture? I don't think you provide this value, but you should. How does it compare to observations, in particular the recent study by Cooley & Christoffersen (2017) could be relevant to determine if this is consistent with observations?

Figure 5: This is a very poor figure and needs changing or removing. In the words of John Turkey: "There is no data that can be displayed in a pie chart, that cannot be displayed better in some other type of chart."

Figure 6: I think a different colour scale would be better. The colours for moulin, crevasse and lake hydrofracture need to stand out from the background colour scale, as do the hatch marks which also get a bit lost. If I understand correctly the total area of the red circles should be ~double that of the blue circles (based on fig 5), which appears to be the case. Can you also include the GPS sites on this figure? I'm interested to see how close the sites are to the supraglacial input locations.

Section 3.2: What exactly are you calibrating in the model – which parameters are being varied to best match the duration and magnitude of speedup events? The results shown in this section are the end result after the calibration (e.g. fig 7 & 8)!? But then that is no different from the 2012 results in section 3.3 (e.g. fig 9)!?

line 11 & elsewhere: "RACMO2" or "RACMO2.3" (from page 3 line 26)

[Figure]

Figure 7: could label on which of the automatic weather stations is displayed (e.g. AWS5 is shown in (b) etc.). The same for Fig 8 & 9.

line 24 & elsewhere: It is standard to separate out multiple units using $\,$ in latex, e.g. $\mathrm{m}\,\mathrm{yr}^{-1}$.

line 7-8: "In this section, ..." This sentence is poor and needs restructuring.

line 16-17: "The impact ... where the water ... was not input to the base." Where was the water put instead? Removed? Drained to the bed less suddenly!? Your sentence appears incomplete, or another sentence to explain is needed.

line 18: "The impact ... was negligible." This surprises me. So velocities at the drainage locations (or just downstream) are unaffected by rapid water input!? Or are you just referring to velocities at the GPS sites (which may not be close to the drainage input locations)?

line 20: "... calibration runs, ..." here I take "calibration runs" to mean the baseline runs after calibration (i.e. Figs 7 & 8).

line 6-7: "... elevated velocities ..." I don't see this in figure 9(a) – two slight blips perhaps!?

line 24-25: "... dependent on melt input." But there appears to be no or little surface runoff at this time (day 238-242)!?

Figure 11: Vertical axis should say "velocity". The figure may be better if the horizontal

axis had a scale as well, e.g. distance from terminus/ice-margin or elevation. Could you also have a vertical dashed line on the graph indicating the equilibrium line.

line 12: "qualitatively" presumably the maximum extent of channelization can be quantified.

line 16-22: What year where the tracers injected?

Figure 13: Are the 3 red dots all L1? These moulin injection sites do not appear to be moulin locations in your model!? I'm comparing to Figure 6!?

lines 1-2: "... there is a brief period ..." Is there? Is this right at the end, after day 250? It's hardly worth mentioning is it!?

Figure 14: A sheet-dominated to channel-dominated drainage "switch" seems only to occur with the 2011x4 results. This is a significant difference with the other simulations and worth discussing in the text. Although, the "switch" does not seem to noticeably effect the ice velocities.

line 3: typo "computational imposed limitations" computationally!?

line 3: typo "... choice ice sheet ..." "... choice of ice sheet ..."

lines 9-10: "... likely to lead to stresses assumed negligible ..." needs rewording

line 13-14: "The atmospheric pressure ..." I'm not sure what exactly you mean by this sentence can you explain further. Is this something that can be easily checked?

line 15: units

line 15: typo "Similarly to the study ..." "Similar to the study ..."

line 16: "The parameter values . . . in Banwell . . ." state their values to help comparison.

lines 27-28: "The initial rapid delivery . . . of water to the bed . . . are not observed to have an effect in [on!?] modelled velocities." Is this just at the GPS sites or everywhere, even close to the drainage locations!?

line 6: ". . . a good match . . ." good might be too strong, perhaps "reasonable".

line 11: "higher order" "higher-order"

line 29: ". . . in line with observations by Hoffman . . ." in line with "findings"!?

line 24: "GriS" "GrIS"

line 31: "Proceedings of a workshop held at Seattle . . ." seems vague, I'm not sure how the journal will view this type of reference.

---

## Referee Comment (RC2) · Anonymous Referee #2 · 9 Dec 2017

This study couples together a surface drainage, ice flow and subglacial hydrology model to examine ice dynamics in the Russell Glacier catchment in Greenland. It's nice to see efforts to couple models like these together as, on their own, it is difficult to fully understand the system. Also, this is a good location to apply this type of study since I think more in situ data has been gathered here than anywhere else in Greenland. In particular, I thought the scenarios of increased melt input were helpful for assessing the dynamic impacts of basal hydrology. The authors argue that the primary output from their model is a confirmation of the hypothesis for an 'Alpine'-like drainage network at the margin of the Greenland Ice Sheet where an initial summer speed-up is followed by deceleration as efficient channel networks develop.

There was a thorough discussion in the paper about the different aspects of the model

and the tests that have been carried out. However, improvements can be made, particularly in the very strong support of the 'Alpine' drainage model, which is really only applicable to the very margin, while the model covers areas much further inland. Also I caution the authors against support of hypotheses applied to the whole of Greenland when Russell Glacier is a fairly slow-moving land-terminating glacier, unlike many of the regions where much of the mass loss is occurring by calving processes at tidewater glaciers. On the whole, I think this paper would be improved by toning down and clarifying some of the arguments as I detail below.

Major points:

The overwhelming argument in the paper is that water gets to the bed in the spring, grows channels and that causes ice deceleration. This is reported as the 'Alpine' model of drainage and, indeed, both the GPS data and the model outputs from this study support that, where the Greenland ice sheet is similar to an Alpine glacier (i.e. near the margin where lots of water gets to the bed and, most importantly, surface slopes are steeper), it does act similar to an Alpine system. However, when you go further inland, both the GPS data and the model in this paper support the argument that has been made in other papers (e.g. Meierbachtol et al, 2013, Andrews et al., 2014, Dow et al, 2015, none of which, by the way, were cited or discussed), that the shallow surface slope prevents large, efficient channels from forming and therefore more water input means higher velocity. This is not the Alpine drainage model. I strongly argue that the authors should recognize that their model outputs show this. The authors do discuss this inland acceleration within the main text but it is not mentioned in either the abstract or conclusion. This acceleration, particularly in the cases of increased melt input, is one of the more important outcomes of the modeling exercise and should therefore be highlighted.

Similarly, the role of channels in the drainage system is something that should be more carefully discussed by the authors. While subglacial channels can grow at high elevations under the Greenland Ice Sheet, it doesn't really matter until they start poaching

water from the distributed system and therefore increase the system effective pressure i.e. presence of channels does not equal efficiency. Unless the surface slopes of the interior steepen considerable, there is no mechanism to efficiently remove water as the hydraulic gradient in channels will not be steep enough and therefore more water will mean faster velocities. So on page 21 where you report channels growing up to 50 km from the margin, that's fine, but these don't appear to be having much impact on the velocity, which is what you're interested in. That should be pointed out and the arguments that more channels inland will mean a velocity decrease, rephrased.

I think the authors need to less strongly argue that the model outputs fit the GPS velocity data well. For example on PG24 13-14 it is stated: 'many of the features observed in the GPS…are captured in the modeled velocities'. On many of the plots in Figures 7 and 8, the GPS velocity is slowing down while the model is speeding up (e.g. 2011 Site 4 day ~203; 2011 site 5 day ~195), and there are few places where I think the model and the GPS would have a good correlation on a day-to-day basis. This doesn't mean that the model outputs aren't useful and I'm not expecting the authors to do any more model runs. However, merely to be more careful with their language. For example, the authors could say: a reasonable fit between seasonal patterns from GPS velocities and modeled velocities.

Line-by-line comments:

PG1

1: can you specify what you mean by margin. How far inland does that go? This doesn't necessarily have to be in the abstract but should be noted somewhere in the main text.

1-2: The first line of the abstract is misleading since you then immediately say that meltwater also leads to deceleration.

PG2

8: where are these measurements?...accumulation or ablation zone? This is confusing following on from the previous sentence.

13: references needed at the end of this sentence e.g. Leeson et al, 2015

23: what do you mean by 'recent hydrological models'?

24: what feedback?

25: 'comparison to ice surface velocity measurements'

27: 'coupled ice dynamics/hydrology models'

29 (and 31): what do you mean by 'necessary elements'? In the context of modeling this is confusing since some of the models you discuss include finite element grids.

30: to drive the subglacial hydrology model?

31-31: repetition of 'different'

32: 'no model to date has included the full spectrum'. This suggests that you do include the full spectrum. From your description of the surface hydrology model it doesn't seem to me like there is anything different from the previous applications of this model. If you have adapted it, you should specify the main changes and why you think this represents the full spectrum.

PG3

1: 'This coupled model...'

18+20: repetition of 90 m resolution topography usage

31: can you specify which boundary conditions you are referring to here.

PG 5

3: why the assumption of -5 degrees C?

4: which Landsat8 image?

14: repetition of 'in'

16: the first sentence is repetition of the 90m resolution for the third time

18: missing reference

20: define what you mean by a fracture area criterion. Also is this fast-drainage dataset driven by that fracture criterion checked against the satellite imagery record of lake drainages?

PG6

1: is that basal no flux condition accurate? Particularly on your upper boundary I would expect some basal water flux into your domain.

6: $K_s$ here but $K$ in Eq 1

10: Eq2 – what is $h_{cav}$?

11 (and 16): $U_b$ is the basal sliding speed but $u_b$ is the basal velocity? What is the difference?

16: how do you calculate basal drag? Also for $u_b$ what is b in the (x,y,b) dimensions? You should also specify what v is.

PG7

3: what do you mean by incipient channel width?

5: what is $x_c$? If this is along-channel distance, what is the difference between this and r?

11-14: How did you choose the % of cells to change and the value to change them to? Did you sensitivity test this? For this whole paragraph you need to include units.

16: what is k?

PG9

6: Perhaps the removal of negative effective pressure is why you aren't replicating the higher upstream GPS velocities.

PG10

3: unclear what you mean by an internal water table. References needed here.

PG11

18: why do you say 'not shown' for the sensitivity analysis? What about section 3.3?

10-12: I don't understand the sentence beginning 'Water routing...'. L10 say 70% routed into crevasses, L11 says 50%.

PG13

3-4: I find it confusing that the surface input is not varied. What do you mean by this?

5: which plots?

7: are you saying because sub-daily variability is subdued, you've averaged to daily? Otherwise this doesn't make sense following the previous sentence.

9: which model output?

PG 14

3-5: I'm afraid it doesn't look like a good fit to me at all. There's a lot of variability in the GPS that aren't captured in your model outputs and events in the model that doesn't appear at all in the GPS. I think you have to have to be much more upfront about these poor fits.

PG16

3: how near the ice sheet margin?

4: low velocities through the summer melt season? Why is there no spring event at the beginning? This is a key of the 'Alpine' hydrology theory so it's worrying if the model doesn't produce that. If you are getting no spring event its possible your overwinter basal water pressures are too low. This is also possibly why your summer deceleration events are not below the winter mean velocity Overwinter the basal water pressure should be at overburden pretty much everywhere so any water input in the spring will cause a velocity increase. It would be good to include the overwinter steady state ice flux/basal water pressure diagrams, perhaps in supplementary material.

17-18: 'In 2009, site S6 shows a gradual...' do you mean the model or the data? Here and elsewhere be careful to specify.

33: Why qualitatively? A correlation should definitely be quantitative.

PG 17

7-8: 'In this section...' awkward sentence.

11: units! Please check the rest of the manuscript to make sure you include units for all of your numbers.

18: I don't understand this. Where does the water go? Also there should definitely be a velocity response (albeit a short one) to a lake drainage event. What is your model timestep by the way?

23: 50% of the initial nodes? What does that mean? Also why these % numbers?

30: how much was englacial storage increased and reduced?

33: day 2015?

PG18

1: I'm getting confused what the difference is between the calibration and the validation runs.

[Figure]

14: accelerating rate of speed-up is poor phrasing

15-16: modeled velocities decrease during periods of slower flow. The sentence is circular and doesn't make sense.

PG21

2: Why does figure 12 compare 2009 with 2011 x 4? Why not 2011 and 2011x4? From this figure it seems like melt increase has a significant impact on the velocity of most of your domain, just not at the very margin. This seems to be understated in your arguments, which are more focused on confirming a model that suggests more water = velocity decrease.

15: the fact that you have channels forming at higher elevations but still get speedup tells you a lot about the system that you haven't discussed. This feeds into one of my major comments about the presence of channels not necessarily facilitating efficiency. It depends how much water those channels remove from the distributed system and therefore the change in effective pressure.

PG25

10: a high aspect ratio of what?

16: cite Schoof et al (2014), The Cryosphere.

PG26

3: 'affected'

24: what is this constant sheet height scale? How does this link to the previous sentence?

27-32: these ideas have been around for a while so you need to reference this paragraph (e.g. Das et al, 2008, Doyle et al, 2013, Dow et al, 2015, Banwell et al, 2016). Also GPS data show that lake drainage have a large impact on ice velocities, just in the

short term. What this means is that your model outputs are not valid for this particular lake hydrofracture example and therefore you have to be careful using them to make strong arguments.

PG 27

13: low aspect ratio and low sliding ratio. Ratios of what?

PG 28

18 (+PG30 6-7): This depends on what you class as the margin. This seems to only apply to within ~40km of the terminus.

PG29

1-8: You're model doesn't support the assertion that channels further inland will cause slowdown. Even though you have more channels, there is still velocity increase at your upper site.

11-12: why would winter velocity decrease? This doesn't make sense if velocities are faster in the higher melt scenarios implying widespread distributed drainage. And what do you mean by offset?

Table 1:

-Ks is called sheet hydraulic conductivity in the text

-is K the hydraulic conductivity for the channel or the sheet? And do you mean m sˆ-1? 2m sˆ-1 is really fast!

- a critical layer depth of 1 m seems high given the basal bumps are 0.1 m high.

- what is the difference between Am and Sm?

Table 2:

- I don't think defining the number of seconds in a year is necessary.

Fig 1: caption needs a date for the satellite image

Fig 3: I don't really understand what this schematic is showing. It's not immediately obvious.

Fig 7/8: An indication of the elevation for each site would be useful here

Fig 14: It's really hard to tell the difference between the sheet and channel curves

---

## Author Comment (AC1) · 25 Jan 2018

We thank both reviewers for their diligence, and their many constructive comments.

Our response is structured as: our reply to reviewer 1, our reply to reviewer 2, and then our updated manuscript with changes highlighted.

**Response to Referee 1.**

We thank the referee for their thorough and positive review. In this document, the referee's comments are given in bold, and our response in normal text. Changes made to the manuscript are given in italics.

This is a comprehensive study that models seasonal ice surface velocities at a land- terminating glacier catchment in Greenland. The work pulls together other published component studies by the authors, including a model of supraglacial meltwater path- ways that drain through moulins into a subglacial hydrology model, which is coupled to an ice-flow model whose sliding parameters have been determined from an inversion using observed winter velocities. It is a challenging task to bring together many mod- elling and observational components and the results show some success. The paper raises many interesting points, including a comparison of results using the Weertman and Schoof sliding laws.

The differences between GPS velocities and modelled velocities highlight areas where future improvements can be made in model development. For example, it is interesting to see some similar challenges in Pimentel et al., Ann. Glac., 2017, e.g. difficulty in capturing early/pre-season speed-up and tendency in the model (with Schoof sliding law) for short-term speed-up events rather than more prolonged enhanced velocities.

In summary I do not have any major problems with the work and think it an interesting and valuable contribution that should be published. I list some minor issues and points of clarification that I would like to see addressed/corrected before final publication.

List of comments: Page 1

**Title: The work is on one particular catchment of the Greenland Ice Sheet, rather than the entire Ice Sheet, this is not clear in the title.**

Although we apply the model to Russell Glacier area, we believe the main contribution from this paper is broader. We show numerically that a multicomponent model including a distributed/channelized subglacial drainage component can be used to recreate ice velocities to a first order. This should be widely applicable beyond the study domain. However, we have amended the title to better reflect the nature of our study.

**line 5: The acronym 'GrIS' is not defined**

We use Greenland Ice Sheet here, and introduce the acronym in the first line of the introduction.

lines 8-10: Do we need further evidence to support that subglacial develop analogous to alpine glaciers and that models need distributed and channelized? This seems quite well established now in the field. Do you want this in the abstract as the key finding of this paper? At the very least 'support' and 'supports' should be changed to 'further support' and 'further supports'.

The evidence for distributed and channelized system of subglacial hydrology beneath the GrIS is mainly indirect: boreholes, dye tracing, and patterns of GPS measurements. However, no one has

shown that this model can reproduce ice velocities (that we are aware of). We think this work uniquely supports it, in that we show numerically that this theory can begin to be applied to recreate and quantitatively predict velocity patterns (with the acknowledgement that there's a lot of work left). We modify the first 'support' to read 'numerically support'

**line 11: 'slow down' of what?**

We add 'of ice velocities'

**line 16: The acronym 'GrIS' is not defined**

Fixed

**line 16: 'margin' should be land-terminating margin, because obviously marine terminating has other additional important influences.**

We have expanded this sentence. While hydrological forcing is the primary driving force in landterminating sectors, we do want the reader to have it in mind for marine-terminating sectors, since we suspect it is non-negligible there.

Surface meltwater draining into the subglacial system drives seasonal acceleration of ice velocities at land-terminating sectors of the Greenland Ice Sheet margin. It may also be a key factor for marine-terminating sectors (Howat et al., 2010; Sole et al., 2011; Moon et al., 2014)

**Page 2**

**lines 12-13: Poor sentence, needs rewriting to make clearer.**

Modified it to read:

As melt season intensity continues to increase, it remains unclear how ice velocities will be forced by water input at higher elevations where ice thickness is greater, and whether patterns of water input at higher elevations will change (Leeson et al., 2015; Poinar et al., 2015; Cooley and Christoffersen, 2017).

**lines 21-23: Presumably just talking about GrIS here.**

Yes, we think this follows from the last sentence, but we have added 'of the GrIS' to clarify.

**line 30: typo "... to the drive the ..."**

deleted the first 'the'

**line 32: I'm not quite sure what you mean by "full spectrum of supraglacial drainage pathways"**

We have modified the sentence to read:

However, no model including drainage via all of crevasses, moulins, and lake hydrofracture has been used to force a subglacial hydrology model to date.

**line 34: How is it similar? You don't seem to have components in the same sense as Arnold et al (1998) and Flowers & Clarke (2002)!?**

We have components in the sense of including surface hydrology, subglacial hydrology, and ice dynamics. Although its true that our model setup isn't exactly the same, we think it's worth acknowledging these efforts which have tried to do something along the same lines in 2D.

line 29: typo "... with with ..."

Fixed

**line 30: A better reference is needed for context of Greenland melt intensity**

Here we are highlighting the similarity of this to our previous work and have reworded the sentence to make this clearer:

Following Koziol(2017), we use these three years as representative of summers with average, elevated, and extreme melt intensity respectively.

**line 31: "... to determine crevasse locations ...." How? Perhaps reference later section at end of sentence.**

This is discussed later in the same section.

**Page 4**

**Figure 1: typo "loactions"**

Fixed

**Page 5**

**line 5: "A threshold value of 145kPa is selected as optimal." Optimal in the sense that it provided the best match to Landsat 8 image?**

Yes, we think this follows from the previous sentence, but have reworded the sentence to make this clearer:

A threshold value of 145 kPa gave the best visual match.

**line 10: "A small number of moulins . . ." could quantify percentage.**

We didn't note the number when we processed the data set since it was only a very small number of moulins. We leave it as it is since we don't think there will be any benefit for the readers for us to determine it, given such moulins would drain negligible amounts of water as stated.

**line 18 & 20: ? in citations**

Fixed

line 21 & throughout: "surface to bed" "surface-to-bed"

updated

line 6: "K\_s" is "K" in equation (1) and table 1, so I think this should be "K" for hydraulic conductivity, whereas you have "K\_s" to denote "sheet flux coefficient" in the table. However, this K\_s does not appear in the text/equations! Hydraulic conductivity is given a value of 2 in the table, which does not match the description in the text!

Yes, this is clearly a mistake. K in Eq 1 should be K\_s. K is deleted.

line 5-7: rho\_w and p\_w are not defined (rho\_w is defined later on line 13).

Fixed.

line 19 & throughout: use \max and \min in latex

Fixed

line 19-20: N\_0 a value is not provided in Table 1.

Fixed

line 6: Presumably "R" is a rate and is a source term from the SRLF?

Exactly. We update the text to state that it is a rate.

**line 10: You take the derivative of Eqn (9) wrt time as a term in Eqn (8). But is Eqn (9) constant!? Or is A\_m a function of time?**

A\_m is constant. The numerators of Eqn (9) are p\_w (water pressure), which is a function of time.

**line 12: "The englacial storage parameter . . ." is this sigma – the "englacial void fraction"!?**

Yes, this is an inconsistency. We have updated it to read englacial void fraction. We move P7L 11-18 to the start of section 3.2

**lines 12-17: units are missing when giving values of conductivity. line 16: "kh^2" should be capital K "Kh^2"**

Following on a previous answer,  $K \rightarrow K_s$ . Included units.

**Page 9**

line 7: "... a small regularization constant" – so this means the zero in  $N_+ = max(N,0)$ , is not actually zero!? Is this constant different from "N\_0" used earlier in equation (4)? What value do

**you use for this regularization constant?**

We've switched the notation here so that N\_+ -> N\_sl. We've amended the text to read:

and regularized with a small constant (\$10^2\$ \unit{Pa}).

**line 16: "... weighted square of the difference of squares ..."!? This doesn't match equation (13) so I assume this should be square of the differences, rather than difference of squares!?**

Yes, this is an error. We've updated the text.

**line 19: please provide your chosen values for gamma\_1 and gamma\_2**

Here we are discussing the model itself. We have added the following sentence to the beginning of the section workflow, P11L10:

(see Koziol and Arnold (2017) for details of inversions, which are done over the same study area but with a resolution of 500 m)

**line 18-22: How was the weighting function chosen?**

We update the last line of the paragraph to read:

The inverse of reported errors of surface velocities are used as weights.

**line 18-22: U\_obs(x,y) – are these just the 7 GPS sites? So the weighting function is 1 at these locations and zero elsewhere!? Ok, I just checked back to Section 2.1, so U\_obs is the winter mean from MEaSURE. And the weights are from the provided error estimates from MEaSURE!?**

Exactly. P11L5 states 'All linear inversions are run using mean winter velocities from 2009...' Here we are focused on describing the model.

**line 18-22: Why isn't "U\_s" also written "U\_s(x,y)"?**

Updated.

**line 18-22: It would be clearer if you explicitly said that U\_s is a function of \alpha and**

**you seek to minimize J with respect to \alpha. If I have understood correctly?**

We never explicitly define what an inversion is in this section, since we figured they are fairly standard in glaciology. However, we modify the text to be more explicit:

The inverse of reported errors of surface velocities are used as weights. Modelled surface velocities depend on the control parameter via the sliding law. The inversion procedure minimizes the cost function with respect to the control parameter.

**line 24: An odd/abrupt start to this subsection**

We have changed the first sentence to read:

The SRLF model is used to determine supraglacial input rates to the subglacial system.

**line 25: There is englacial storage in the subglacial model, but water from the SRLF is not directed to this storage – correct!?**

Yes, we have modified this to read:

For model integration, we assume that water drainage through surface-to-bed connections is strictly vertical, with no horizontal component.

**line 27 "All water entering these cells drains directly to the bed" So water is not stored in the moulins – how does this relate to Eqn (9)!?**

This was imprecise. There is water storage in moulins. We think this new sentence better communicates our point:

All water entering these cells drains into the subglacial hydrology system at that location.

**line 29: "... the water is removed from the model ..." do you actually mean the water vanishes!? Because on page 10 line 2 you say "... all water in crevasse fields drain to bed ...". If I understand correctly the water in the crevassed grid cells ends up draining into a nearby moulin where it then enters the bed.**

The SRLF model is strictly a surface hydrology model. When water enters a cell with a surface to bed connection, it is removed from the model. Since it's no longer on the surface, no further routing needs to occur. We then have to postprocess these results to provide input to the subglacial hydrology model. For moulins/lake hydrofracture, we can put the water into the subglacial system at the same location. Since not every crevasse cell will drain locally to the bed, these require more involved post-processing.

Thus, we have removed the clause ". . . the water is removed from the model . . .", which we think should avoid this confusion.

**line 29: "... and no further routing occurs." It is instantly routed to the corresponding moulin – if I understand your next paragraph correctly!?**

No further routing occurs in the SRLF model. We have to postprocess the results and implement the routing after the SRLF model run. We rearrange the sentence slightly to be clearer.

When water enters a crevassed cell in the SRLF model, no further routing occurs.

Page 10 line 11: "veronoi" "Veronoi"

**Fixed**

**Figure 3: It may be helpful to somehow illustrate the crevasses (crevasse field). You could possibly use hatched grid cells, like in Figure 6. I suppose this is implemented in a finite difference grid, so the curved lines of the internal catchment would actually be straight edged.**

We have shaded the area within the crevasse field, and added a corresponding line in the caption. Hatched lines are not a default option in the open source graphics editor we are using (Inkscape). In a finite difference scheme any curved line is made up of small segments of straight lines, we think this is a negligible detail. A schematic drawing implies that the elements are representative rather than realistic.

**Page 11**

**line 8: "The effective pressures at the end of the . . . simulation . . ." Presumably they've stabilized by then?**

Yes, we see a large initial transient response in the first 50 days or so. We have added a line stating that they reach an approximate steady state by the end of the winter run.

**line 10: "coefficients." State the coefficients, i.e. "coefficients, mu\_a and mu\_b."**

Updated

**line 10: "... the basal water pressures, ... form inputs ..." how are the basal water pressures used? Initial conditions of the subglacial summer simulations!?**

We have modified the next sentence to read:

**The integrated model is then run for the summer melt season **using the end of winter effective** *pressures as an initial condition.**

**line 16: "A parameter search . . . ". I'm not sure what you are getting at with this sentence.**

We have modified this sentence to read:

A parameter search therefore requires performing the inversion using an effective pressure dependent sliding law for each set of parameters tested.

The implication here is that it is costly.

**line 18: "... sensitivity analysis (not shown)." Section 3.3!?**

We have removed this clause as both reviewers commented on it. What we meant for this to communicate is we don't show each of the nearly 200 sensitivity runs, but rather the final calibrated runs.

**Page 12**

Section 3.1: What percentage of your lakes hydrofracture? I don't think you provide this value, but you should. How does it compare to observations, in particular the recent study by Cooley &

**Christoffersen (2017) could be relevant to determine if this is consistent with observations?**

We have amended the text as:

**Approximately 12\% of supraglacial lakes are predicted to hydrofracture. These events drain only a small percentage of surface runoff (1.3\%).**

There are comparisons of the SRLF model results to satellite imagery in Arnold et al. 2015, as well as Koziol et al., 2017. In the latter, we refine the calibration so that ~11% of lakes hydrofracture, and ~38% of lakes drain via channel incision. Although the calibration is for the Paakitsoq region (just north of Jakobshavn Isbrae), the numbers we calibrate to are for SW Greenland, and we expect it to be valid in our present study domain.

**Figure 5: This is a very poor figure and needs changing or removing. In the words of John Turkey: "There is no data that can be displayed in a pie chart, that cannot be displayed better in some other type of chart."**

We have replaced it with a table.

**Page 13**

Figure 6: I think a different colour scale would be better. The colours for moulin, crevasse and lake hydrofracture need to stand out from the background colour scale, as do the hatch marks which also get a bit lost. If I understand correctly the total area of the red circles should be ~double that of the blue circles (based on fig 5), which appears to be the case. Can you also include the GPS sites on this figure? I'm interested to see how close the sites are to the supraglacial input locations.

We put a lot of effort into making this image; the colour scale is the best we found. The hatchmarks are difficult to modify, as it's quite a hack in Matlab to get them to appear. Overall, we think the appearance of the figure is satisfactory, but we have included GPS positions as black triangles as suggested by the reviewer, and updated the caption.

**Section 3.2: What exactly are you calibrating in the model – which parameters are being varied to best match the duration and magnitude of speedup events? The results shown in this section are the end result after the calibration (e.g. fig 7 & 8)!? But then that is no different from the 2012 results in section 3.3 (e.g. fig 9)!?**

We move P7L 11-18 to the start of section 3, and add clarify by adding:

The focus of the calibration is on parameters identified as key to determining the morphology of the subglacial system,  $K_s$  and s.

**line 11 & elsewhere: "RACMO2" or "RACMO2.3" (from page 3 line 26)**

Updated to RACMO2.3 throughout

**Page 14**

Figure 7: could label on which of the automatic weather stations is displayed (e.g. AWS5 is shown in (b) etc.). The same for Fig 8 & 9.

Updated

**Page 16**

line 24 & elsewhere: It is standard to separate out multiple units using \$\,\$ in latex,

**e.g. $\operatorname{\mathbb{yr}}^{-1}$**

Updated

**Page 17**

line 7-8: "In this section, . . . " This sentence is poor and needs restructuring.

Rewrote as:

In this section we focus on the sensitivity of the model to the setup, to \$K\_s\$, and to \$\sigma\$.

line 16-17: "The impact . . . where the water . . . was not input to the base." Where was the water put instead? Removed? Drained to the bed less suddenly!? Your sentence appears incomplete, or another sentence to explain is needed.

We have amended the sentence by adding the clause ', *but removed from the system*.' to the end of the sentence.

line 18: "The impact . . . was negligible." This surprises me. So velocities at the drainage locations (or just downstream) are unaffected by rapid water input!? Or are you just referring to velocities at the GPS sites (which may not be close to the drainage input locations)?

We amend the sentence to state we are referring to the GPS stations.

line 20: "... calibration runs, ..." here I take "calibration runs" to mean the baseline runs after calibration (i.e. Figs 7 & 8).

We change 'calibration' to 'calibrated', which should be a lot clearer.

**Page 18**

line 6-7: "... elevated velocities ..." I don't see this in figure 9(a) - two slight blips perhaps!?

Yes, this is hard to see. We update the text to say ...'slightly elevated velocities...' The model clearly gets the magnitude wrong, however the timing is there.

line 24-25: "... dependent on melt input." But there appears to be no or little surface runoff at this time (day 238-242)!?

Plotting mistake. – the background melt in the figure is for 2012, not for 2011. We have updated this figure.

**Page 21**

**Figure 11: Vertical axis should say "velocity". The figure may be better if the horizontal axis had a scale as well, e.g. distance from terminus/ice-margin or elevation. Could you also have a vertical dashed line on the graph indicating the equilibrium line.**

Updated vertical and horizontal axis. We now show an elevation profile in this figure as well. We decided the figure looked better without the vertical line indicating the equilibrium line.

**line 12: "qualitatively" presumably the maximum extent of channelization can be quantified.**

Yes, we have the data, but the files are cumbersome to work with, and we don't feel there is a significant benefit from quantitatively performing the analysis.

**line 16-22: What year where the tracers injected?**

We amend the paragraph to say:

Dye-tracing experiments by \citet{Chandler2013} were performed in the summers of 2009 to 2011.

and

Based on tracer measurements in 2011

**Figure 13: Are the 3 red dots all L1? These moulin injection sites do not appear to be moulin locations in your model!? I'm comparing to Figure 6!?**

Only the closest dot to the margin is L1. We amend the caption to say:

Two moulin locations are unlabeled for clarity.

Comparing Fig 13 to Fig 6, we agree that L1 does not appear. It may be a crevassed cell though. The first unlabeled site (closer to the margin) would be one of the crevasse cells. Note when our crevasse prediction overlapped with a moulin observation, the moulin was deleted, as it would not receive any water in the model. The second unlabeled injection site (further from the margin), appears in Fig 6. IS39 and L41 have moulins nearby in Fig 6, and L57 does not appear – although there are moulins upstream. There is some discrepancy as observed moulin locations are from 2014, and moulin locations from Chandler et al (2013) are from 2009-2011.

**Page 24**

**lines 1-2: "... there is a brief period ..." Is there? Is this right at the end, after day 250? It's hardly worth mentioning is it!?**

We modify the sentence to be more specific that we are referring to 2011 and 2012.

Figure 14: A sheet-dominated to channel-dominated drainage "switch" seems only to occur with the 2011x4 results. This is a significant difference with the other simulations and worth discussing in the text. Although, the "switch" does not seem to noticeably effect the ice velocities.

We have expanded the discussion of this within the text.

**line 3: typo "computational imposed limitations" computationally!?**

Updated

line 3: typo "... choice ice sheet ..." "... choice of ice sheet ..."

Updated

lines 9-10: "... likely to lead to stresses assumed negligible ...." needs rewording

**We have simplified the whole sentence to:**

Steep surface gradients may lead to stresses assumed negligible by the hybrid formulation in the ice stress balance.

**line 13-14: "The atmospheric pressure . . ." I'm not sure what exactly you mean by this sentence can you explain further. Is this something that can be easily checked?**

Reworded as:

*The atmospheric pressure boundary condition prescribed at the ice sheet margin in the subglacial hydrology model, ...*

This is not something easily checked. However, observations of R-channels exiting glaciers show that they are not always full. We believe it is a reasonable assumption that this could extend some way away from the ice margin, and that these channels could oscillate from partially full (or unpressurized) to full. The model assumes these channels are always full (pressurized).

**line 15: units**

We don't see any numbers on this line, this might belong on the next page where we missed units. **line 15: typo "Similarly to the study . . . " "Similar to the study . . . "**

Updated

**Page 26**

**line 16: "The parameter values . . . in Banwell . . . " state their values to help comparison.**

We have reworded the section as follows to be more clear:

Most model parameters of the integrated model are similar to previous studies applying the subglacial hydrology model (Hewitt, 2013; Banwell et al., 2016). The most significant parameter value difference

is the sheet flux coefficient (Ks). The primary value of 10-2 Pa-1s-1 in our study is greater than the value of 10-5 Pa-1s-1 used in Banwell et al. (2016). The parameter values for the model reported in Banwell et al. (2016),...

**lines 27-28: "The initial rapid delivery . . . of water to the bed . . . are not observed to have an effect in [on!?] modelled velocities." Is this just at the GPS sites or everywhere, even close to the drainage locations!?**

We have re-worded this sentence to make it clearer that we are discussing the broader impact on ice velocity, rather than any short-term, localised effect.

The initial rapid delivery of a large volume of water to the bed during individual lake hydrofracture events is not observed to have a lasting or widespread effect on modelled velocities.

line 6: "... a good match ... " good might be too strong, perhaps "reasonable".

Updated

**line 11: "higher order" "higher-order"**

updated

**line 29: "... in line with observations by Hoffman ...." in line with "findings"!?**

Their work is based on GPS measurements, so we think 'observations' is the correct word to use.

line 24: "GriS" "GrIS"

Updated

**line 31: "Proceedings of a workshop held at Seattle . . ." seems vague, I'm not sure how the journal will view this type of reference.**

We update the reference to:

Raymond, C. and Nolan, M.: Drainage of a glacial lake through an ice spillway, Intl. Assoc. Hydrol. Sci. Publ., 264, 199–210, http://iahs. info/uploads/dms/iahs\_264\_0199.pdf, 2000.

which is how it was cited in our Journal of Glaciology paper (Koziol et al., (2017)). This paper has been cited in Kingslake et al. (2015, JGlac), and Jarosch and Gudmusson (2012, TC) as well.

**Response to Referee 2.**

We thank the referee for their thorough review. We accept that certain aspects of the text could be reworded with a different emphasis to better reflect our findings and their implications. In this document, the referee's comments are given in bold, and our response in normal text. Changes made to the manuscript are given in italics.

This study couples together a surface drainage, ice flow and subglacial hydrology model to examine ice dynamics in the Russell Glacier catchment in Greenland. It's nice to see efforts to couple models like these together as, on their own, it is difficult to fully understand the system. Also, this is a good location to apply this type of study since I think more in situ data has been gathered here than anywhere else in Green- land. In particular, I thought the scenarios of increased melt input were helpful for assessing the dynamic impacts of basal hydrology. The authors argue that the primary output from their model is a confirmation of the hypothesis for an 'Alpine'-like drainage network at the margin of the Greenland Ice Sheet where an initial summer speed-up is followed by deceleration as efficient channel networks develop.

There was a thorough discussion in the paper about the different aspects of the model C1 and the tests that have been carried out. However, improvements can be made, par- ticularly in the very strong support of the 'Alpine' drainage model, which is really only applicable to the very margin, while the model covers areas much further inland. Also I caution the authors against support of hypotheses applied to the whole of Greenland when Russell Glacier is a fairly slow-moving land-terminating glacier, unlike many of the regions where much of the mass loss is occurring by calving processes at tidewater glaciers. On the whole, I think this paper would be improved by toning down and clarifying some of the arguments as I detail below.

**Major points:**

The overwhelming argument in the paper is that water gets to the bed in the spring, grows channels and that causes ice deceleration. This is reported as the 'Alpine' model of drainage and, indeed, both the GPS data and the model outputs from this study support that, where the Greenland ice sheet is similar to an Alpine glacier (i.e. near the margin where lots of water gets to the bed and, most importantly, surface slopes are steeper), it does act similar to an Alpine system. However, when you go further inland, both the GPS data and the model in this paper support the argument that has been made in other papers (e.g. Meierbachtol et al, 2013, Andrews et al., 2014, Dow et al, 2015, none of which, by the way, were cited or discussed), that the shallow surface slope prevents large, efficient channels from forming and therefore more water input means higher velocity. This is not the Alpine drainage model. I strongly argue that the authors should recognize that their model outputs show this. The authors do discuss this inland acceleration within the main text but it is not mentioned in either the abstract or conclusion. This acceleration, particularly in the cases of increased melt input, is one of the more important outcomes of the modeling exercise and should therefore be highlighted.

We think the article communicates a broader point. The main contribution from our perspective lays with showing that a multicomponent model incorporating both distributed and channelized systems of subglacial hydrology can be used to recreate GPS velocities to a first order. There is indirect evidence for the existence of a distributed/channelized system beneath the GrIS from observation of boreholes, dye tracing, and the GPS velocity patterns, but to date, no model has been developed including all these elements which is able to recreate surface velocity measurements. The model we develop in this paper

provides numerical support for this understanding of subglacial hydrology beneath the GrIS, as well support for the explicit description of the model components (i.e. the equations used). Since the model works to a first order, we also aimed to investigate what ice velocities may look like in the future through a series of simple tests. The model predicts that ice velocities will scale with summer melt input (which we state in the abstract and conclusion via 'strong summer velocity scaling').

The quantity of interest in order to understand possible dynamic changes affecting ice sheet mass balance is annual average ice velocities, rather than summer average velocities. Since we do not model winter velocities, to make this link, we rely on trends observed in GPS velocities. Previous literature shows that in the upper ablation zone of the Russell Glacier Area, there is an inverse relationship between summer velocity and annual velocity (e.g. Figure 7, Van de Wal et al., 2015). As melt input increases, annual velocities decrease at a faster rate than average summer velocities increase. This is explained to be due to the effect of channelization on winter velocities (an argument made by others, including Sole et al. 2013). Our model predicts both faster summer velocities, and increased rates of channelization for higher melt scenarios. Hence, we make the assumption that current observed trends will continue, and investigate the limit of this behaviour, which is when winter velocities will hit their lower bound. Although the assumption we make is not constrained by our model output, and has uncertainty in it due to the points raised by the reviewer, we think it is quite reasonable.

It is evident from the equations that thick ice and shallow slopes inhibit channel growth. However, it is not clear to us that our system shows a limit where increased water input results in increased annual ice velocities, as we do not model winter ice flow. However, we now include this discussion, and the references mentioned. In particular, we also discuss the model implications in the limit of summer velocities increasing but winter velocities remaining constant.

Most of the understanding of subglacial hydrology we have comes from alpine glaciers. This includes the concepts of distributed and channelized systems, as well as the equations. An alpine drainage system encompasses these elements, and the term has been widely used in the literature when discussing the behaviour of the Greenland Ice Sheet.

In order to better emphasize the points raised above and also to reflect the reviewer's comments, we have made the following changes.

We better highlight the link between average summer ice velocities and melt season intensity in the abstract.

**P1 L11-14 (Abstract)**

This suggests current trends of decadal timescale slow-down of ice velocities in the ablation zone may continue in the near future. The model results also show a strong scaling between average summer velocities and melt season intensity, particularly in the upper ablation area. Assuming winter velocities are not impacted by channelization, our model suggests an upper bound of a 25% increase in annual surface velocities as surface melt increases to 4x present levels.

We delete this sentence.

P28 L18. The model results also support the hypothesis that the margin of the GrIS is controlled by

subglacial hydrology in a manner similar to alpine glaciers.

We then move lines P28 L12-17 to the end of the section (P29 L23), and insert this paragraph in its place.

The existence of channelized and distributed systems beneath the GrIS is inferred indirectly through borehole observations, dye tracing experiments, and patterns of GPS velocities, building on extensive observations and theoretical developments derived from studies on alpine glaciers. The key result of this paper is to provide numerical support for the understanding of subglacial hydrology of the GrIS, based on theories derived from studies of alpine glaciers, as well support for the explicit description of the model components we include (i.e. the equations used). We show that these theories can quantitatively reproduce measurements to a first order, and in the sense of our validation, predict ice velocities. This builds on previous work which shows that this understanding can be used to reproduce idealized seasonal patterns of ice velocity \citep(Pimentel2011, Colgan2012, Hewitt2013), as well as effective pressures in line with ice velocities \citep{Werder2013,Fleurian2016}.

We have also expanded the discussion on P29 of velocity sensitivity to melt considerably.

However, as channelization increases up-ice in our model, we do not see a marked impact on model velocities. Model velocities at the higher GPS stations in model runs 2011x2 and 2011x4 both show a similar pattern to 2011, with a higher magnitude of ice flow during speedup events. We do not observe a shift in velocity patterns towards that of lower GPS stations, with acceleration early in the melt season transitioning to deceleration in the latter part of the melt season. This suggests that channelization may have a more limited impact on annual velocities in the accumulation zone. The magnitude of any impact is unresolved by our model. In particular, although there are periods when velocities in the 2011x4 run are lower than 2011x2 and 2011, the magnitude of this decrease is bounded. This is a limitation of the model, which is only able to decrease velocities nominally below the initial winter values. This also has the consequence that we are unable to model the winter season accurately.

The key question for the longer term response of the ice sheet to increased melt is whether the potential summer increase in velocity due to increased melt will outweigh any late summer and winter decrease due to the evolution of a more efficient system under higher melt conditions. While observations show long-term decreases in ice velocities in the lower ablation zone \citep{Stevens2016, Tedstone2015, VandeWal2015}, this question remains unresolved at higher elevations. Although we cannot directly predict annual velocities with the model presented in this study, we can investigate how annual velocities may change at the limits of winter behaviour.

One limit of winter velocities is that integrated ice flow over the winter decreases faster than integrated ice flow over the summer. This is observed in GPS measurements in the upper ablation zone in the Russell Glacier region by \citet{VandeWal2015}. Under this limit, channelization has a similar impact at high elevations as in the ablation zone. Velocity measurements near the vicinity of a lake hydrofracture at approximately 1450 \unit{m} elevation suggest that channelization occurs even at high elevations \citep{Bartholomew2012, Nienow2017}. Winter velocities in this limit will decrease until they hit a lower bound where flow is purely deformational, with no contribution from basal

sliding. The maximum increase in mean summer velocities is approximately 60 \unit{m yr \{-1}}, at GPS stations 4 and 5 between the 2009 and 2011x4 melt scenarios. Assuming a winter velocity of 100 \unit{m yr \{-1}} and an 7 month-long winter, the summer increase predicted by the model would compensate for a possible reduction in winter velocity to around 60 \unit{m yr \{-1}}. This approaches the lower bound for winter velocity suggested by borehole measurements showing that internal deformation accounts for 25-50\% of the total ice velocity in the Paakitsoq region of western Greenland \citep{Ryser2014}. Climate model predictions suggest surface runoff rates quadrupling from present levels circa 2100 \citep{Shannon2013, VanAngelen2013}.

The second limit occurs if winter velocities at higher elevations are not impacted by channelization, and summer velocities dominate the annual signal. Arguments that thick ice and shallow surface slopes inhibit channel growth at high elevations favour this limit of behaviour \citep{Meierbachtol2013,Dow2014}, as do observations suggesting limited changes in the efficiency of the channelized system \citep{Andrews2014}. Under this scenario, a change of mean summer velocity of 110 \unit{m yr/{-1}} to 170 \unit{m yr/{-1}} with winter velocities remaining constant at 100 \unit{m yr/{-1}} would result in mean annual velocities increasing from 104 \unit{m yr/{-1}} to 129 \unit{m yr/{-1}} between the 2009 and 2011x4 simulations. Under this scenario, annual velocities would increase by approximately 25\% by approximately 2100, when surface runoff is predicted to quadruple.

Similarly, the role of channels in the drainage system is something that should be more carefully discussed by the authors. While subglacial channels can grow at high elevations under the Greenland Ice Sheet, it doesn't really matter until they start poaching water from the distributed system and therefore increase the system effective pressure i.e. presence of channels does not equal efficiency. Unless the surface slopes of the interior steepen considerable, there is no mechanism to efficiently remove water as the hydraulic gradient in channels will not be steep enough and therefore more water will mean faster velocities. So on page 21 where you report channels growing up to 50 km from the margin, that's fine, but these don't appear to be having much impact on the velocity, which is what you're interested in. That should be pointed out and the arguments that more channels inland will mean a velocity decrease, rephrased.

On P21, we are simply discussing the results of the model, not their implications. However, we accept the arguments raised by the referee here, and we have modified some aspects of the discussion and conclusions to reflect this, as discussed in our response to the general comments. Also, we already discuss the idea that the overall scaling of velocity with increasing melt suggests that the growth of channels does not cause a velocity decrease when viewed at the scale of the whole model domain.

I think the authors need to less strongly argue that the model outputs fit the GPS velocity data well. For example on PG24 13-14 it is stated: 'many of the features observed in the GPS. . .are captured in the modeled velocities'. On many of the plots in Figures 7 and 8, the GPS velocity is slowing down while the model is speeding up (e.g. 2011 Site 4 day ~203; 2011 site 5 day ~195), and there are few places where I think the model and the GPS would have a good correlation on a day-to-day basis. This doesn't mean that the model outputs aren't useful and I'm not expecting the authors to do any more model runs. However, merely to be more careful with their language. For example, the authors could say: a reasonable fit between seasonal patterns from GPS

**velocities and modeled velocities.**

We have tried, where appropriate, to be more careful when describing the level of agreement between the model results and observed velocity. In particularly in Sections 3.2 and 3.4 we are careful to discuss both areas of agreement between the model and observations, but also areas of mis-match.

**Line-by-line comments:**

**PG1**

**1: can you specify what you mean by margin. How far inland does that go? This doesn't necessarily have to be in the abstract but should be noted somewhere in the main text.**

'Margin' isn't meant as a technical term, and has been used in its general English meaning of the edge or border, in line with other papers in the literature.

**1-2: The first line of the abstract is misleading since you then immediately say that meltwater also leads to deceleration.**

We state ice flows faster during the summer. The pattern of ice flow shows acceleration early in the summer (du/dt > 0), and deceleration in the latter part of the summer (du/dt

[revised manuscript text omitted]

---

## Author Response (AR2)

We thank the editor for the careful read of our manuscript. Our replies follow.

**p. 1 line 2: unlike stated in the response the abbreviation of (GiIS) is missing here after ‚Greenland Ice Sheet' and should be added.**
Fixed

**p. 2 line 23-24: plural or singular? either its should be '…to an ice flow model…' or '…to ice flow models…'**
Fixed

**p. 5 line 5: there is the degree circle missing between '-5' and 'C'**
Fixed

**p. 6, 7 and 8: some of the symbols (variables, constants) in the text and in the tables 1 and 2 are not italic, but in the equations they are. (e.g. constants g,… exponents n, …, variables/coefficients m, G, S, r, M, R, p, q,…).**
**I have the impression that you wanted to maybe have the constants and exponents (normal) non-italic but in the equations they are italic anyway. So I suggest to make them all consistently italic (at least the same in text, equations and tables).**

We have switched over to italics.

**p. 9 line 27: would 'equally' not be better here than 'identically'.**
We have left this unchanged. The two features are treated identically by the model code; equally in this context has a looser meaning.

**p. 9 line 28: 'crevasse fields' (plural) (as later you always use the plural)**
Fixed

**p. 10 line 12: '…drain INTO its…'**
Fixed

**p. 14 line 13: '…schoof and Budd slinding laws…'**
Fixed

**p. 14 line 333: … velocities…show a gradual…' (plural)**
Fixed

**p. 14 line 333: … measurements show …' (plural)**
Fixed

**p. 17 line 14-16: I agree with the fereree that 'correlation' is not the ideal term for 'qualitative' agreement as it has a set 'quantitative' meaning in statistics. Maybe replace/rephrase to something with agrees/corresponds/…**
We have used 'correspond' here to replace all the uses of correlate.

**p. 17 line 33: there are some comma missing here I think. '…where the water in the lake, when hydrofracture occurs, was not…'**

Fixed

**p. 18 line 8: again add a comma after 'points'**
Fixed

**p. 23 line 4: '…injection (singular) points…'**
Fixed

**p. 26 line 25-26: I think the citation Smithe et al (should be outside the brackets e.g. '…results of the Smith et al. (2017) study.**
Fixed

**p. 27 line 7 'Simulations results show (plural)…'**
Fixed

**p. 28 line 6: delete the 'in' after 'within'**
Fixed

**p. 29 line 13: 'This calculation needs (singular) only be done…'**
Fixed

**p. 29 line 19: I think there is an 'as' missing between 'well' and 'support'.**
Fixed

**p. 30 line 21: style, maybe replace 'has the consequence' by 'implies'**
Fixed

**p. 31 line 5: '…levels TO circa 2100…'**
We have inserted by, rather than to here, as 2100 is a year.

**p. 31 line 7 : 'THE argument …' (singular, as you only give one here)**
Fixed

**p. 31 line 12: to avoid repetition maybe delete the 'approximately' before '2100'.**
Changed approximately to circa, to match our previous useage.

**p. 31 line 25: something wrong in this sentence, do you mean.: '…of high complexity that incorporates a range of processes.'**
Fixed

**p. 32 line 17: I would say 'close to' as otherwise you would have to say 'closer to what????'**
Fixed

**p. 32 line 22-23: I find it rather hard to follow this sentence and would simplify it a bit by simply saying: 'In the second limit, for which winter ….'.**
Fixed

**p. 32 Acknowledgnments: please acknowledge the two referees in the Acknowledgements**

Added

**caption figure 10: maybe be more specific here and extend to:

[revised manuscript text omitted]